# ATP activation of peritubular cells drives testicular sperm transport

David Fleck[1†], Lina Kenzler[1†], Nadine Mundt[1,2], Martin Strauch[3], Naofumi Uesaka[1,4], Robert Moosmann[1], Felicitas Bruentgens[1‡], Annika Missel[5], Artur Mayerhofer[5], Dorit Merhof[3], Jennifer Spehr[1], Marc Spehr[1,2]*

[1]Department of Chemosensation, Institute for Biology II, RWTH Aachen University, Aachen, Germany; [2]Research Training Group 2416 MultiSenses – MultiScales, RWTH Aachen University, Aachen, Germany; [3]Institute of Imaging and Computer Vision, RWTH Aachen University, Aachen, Germany; [4]Department of Cognitive Neurobiology, Tokyo Medical and Dental University, Tokyo, Japan; [5]Biomedical Center Munich (BMC), Cell Biology, Anatomy III, Ludwig-Maximilians-Universität München, Planegg-Martinsried, Germany

*For correspondence:
m.spehr@sensorik.rwth-aachen.de

[†]These authors contributed equally to this work

Present address: [‡] Charité-Universitätsmedizin Berlin, Neuroscience Research Center, Berlin, Germany

Competing interests: The authors declare that no competing interests exist.

**Abstract** Spermatogenesis, the complex process of male germ cell proliferation, differentiation, and maturation, is the basis of male fertility. In the seminiferous tubules of the testes, spermatozoa are constantly generated from spermatogonial stem cells through a stereotyped sequence of mitotic and meiotic divisions. The basic physiological principles, however, that control both maturation and luminal transport of the still immotile spermatozoa within the seminiferous tubules remain poorly, if at all, defined. Here, we show that coordinated contractions of smooth muscle-like testicular peritubular cells provide the propulsive force for luminal sperm transport toward the rete testis. Using a mouse model for in vivo imaging, we describe and quantify spontaneous tubular contractions and show a causal relationship between peritubular $Ca^{2+}$ waves and peristaltic transport. Moreover, we identify P2 receptor-dependent purinergic signaling pathways as physiological triggers of tubular contractions both in vitro and in vivo. When challenged with extracellular ATP, transport of luminal content inside the seminiferous tubules displays stage-dependent directionality. We thus suggest that paracrine purinergic signaling coordinates peristaltic recurrent contractions of the mouse seminiferous tubules to propel immotile spermatozoa to the rete testis.

## Introduction

Spermatogenesis ranks among the most complex, yet least understood developmental processes in postnatal life. Inside the seminiferous tubules, which represent the functional units of the testis, this intricate course of mass cell proliferation and transformation events generates haploid spermatozoa from diploid spermatogonial stem cells. The seminiferous epithelium is composed of Sertoli cells, each intimately associated with ≥30 germ cells at different developmental stages (*Mruk and Cheng, 2004*). Sertoli cells provide the microenvironment critical for spermatogenesis by establishing the blood-testis barrier (*Cheng and Mruk, 2010*), forming the spermatogonial stem cell niche (*Oatley and Brinster, 2012*), and controlling epithelial cyclicity via auto-, para-, and endocrine feedback (*Heindel and Treinen, 1989*). Different types of spermatogonia (type A, intermediate, type B) are localized along the seminiferous tubule basement membrane (*Chiarini-Garcia and Russell, 2001*). Upon detachment, type B spermatogonia enter meiosis as preleptotene spermatocytes. During meiotic divisions and subsequent maturation steps, germ cells progress from primary to secondary spermatocytes and round to elongated spermatids.

**eLife digest** As sperm develop in the testis, the immature cells must make their way through a maze of small tubes known as seminiferous tubules. However, at this stage, the cells do not yet move the long tails that normally allow them to 'swim'; it is therefore unclear how they are able to move through the tubules.

Now, Fleck, Kenzler et al. have showed that, in mice, muscle-like cells within the walls of seminiferous tubules can create waves of contractions that push sperm along. Further experiments were then conducted on cells grown in the laboratory. This revealed that a signaling molecule called ATP orchestrates the moving process by activating a cascade of molecular events that result in contractions. Fleck, Kenzler et al. then harnessed an advanced microscopy technique to demonstrate that this mechanism occurs in living mice. Together, these results provide a better understanding of how sperm mature, which could potentially be relevant for both male infertility and birth control.

Accumulating evidence implicates purinergic signaling in testicular paracrine communication. While the general picture is still incomplete, cell- and stage-specific testicular expression of different purinoceptor isoforms has been reported in Leydig cells (*Antonio et al., 2009*; *Foresta et al., 1996*), Sertoli cells (*Veitinger et al., 2011*), both pre- and postmeiotic germ cells (*Fleck et al., 2016*; *Glass et al., 2001*), testicular peritubular cells (TPCs) (*Walenta et al., 2018*), as well as mature spermatozoa (*Navarro et al., 2011*), albeit with contradictory results. Functionally, several studies have suggested purinergic paracrine control of gonadotropin effects on Leydig and Sertoli cells (*Filippini et al., 1994*; *Gelain et al., 2005*; *Gelain et al., 2003*; *Lalevée et al., 1999*; *Meroni et al., 1998*; *Poletto Chaves et al., 2006*), including steroidogenesis and testosterone/17β-estradiol secretion (*Foresta et al., 1996*; *Rossato et al., 2001*).

Members of the P2 purinoceptor family are activated by extracellular ATP (*Burnstock, 1990*). P2 receptors subdivide into ionotropic P2X (*Bean, 1992*; *Bean and Friel, 1990*) and metabotropic P2Y (*Barnard et al., 1994*) receptors, comprising seven (P2X) and eight (P2Y) isoforms, respectively (*Müller et al., 2020*). All P2X channels display substantial $Ca^{2+}$ permeability and show distinct pharmacological profiles, ligand affinities, and desensitization kinetics (*Khakh and North, 2012*). G-protein-coupled P2Y receptors are sensitive to both ATP and UTP and they form two subgroups that either activate phospholipase C via $G_{\alpha q}/G_{\alpha 11}$ (P2Y1, 2, 4, 6, and 11) or couple to $G_{\alpha i/o}$ (P2Y12, 13, and 14) (*Müller et al., 2020*). Notably, several P2 receptor isoforms affect smooth muscle cell physiology, with P2X1, P2X2, P2X4, P2X7, P2Y1, and P2Y2 acting as the principle subunits (*Burnstock, 2014*). So far, the most prominent role for a specific subunit in reproductive physiology has been attributed to P2X1, which is critical for vas deferens smooth muscle contraction and male fertility (*Mulryan et al., 2000*).

In mice, stimulation-dependent ATP secretion from both Sertoli and germ cells was reported (*Gelain et al., 2005*; *Gelain et al., 2003*) and may itself be under endocrine control (*Gelain et al., 2005*; *Lalevée et al., 1999*). The mechanism(s) of cellular ATP release, however, remain subject to debate. ATP secretion via exocytotic release (*Bodin and Burnstock, 2001*; *Zhang et al., 2007*) has been proposed. Alternative ATP release pathways include connexin/pannexin hemichannels (*Bao et al., 2004*; *Cotrina et al., 1998*), transporters (*Lohman et al., 2012*), voltage-gated (*Taruno et al., 2013*) or large-conductance anion (*Bell et al., 2003*) channels, or even P2X7 receptors (*Pellegatti et al., 2005*; *Suadicani et al., 2006*).

Along the seminiferous epithelium, spermatogenesis has been conceptualized by attribution of sequential cellular 'stages' (*Figure 1A*), which progress through coordinated cycles (*Hess and De Franca, 2008*; *Russell, 1990*). First initiated in mice about 7 days postpartum (*Kolasa et al., 2012*), each spermatogenic cycle comprises 12 stages and lasts 8.7 days (*Hermo et al., 2010*). After approximately 39 days (4.5 cycle repetitions), spermatogenesis completes with the release of immotile spermatozoa from the seminiferous epithelium into the lumen of the tubule (spermiation). Once detached from the Sertoli cells, sperm must be transported to the *rete testis* and epididymis for final maturation. Precisely regulated tubular transport mechanisms are, thus, imperative for reproduction.

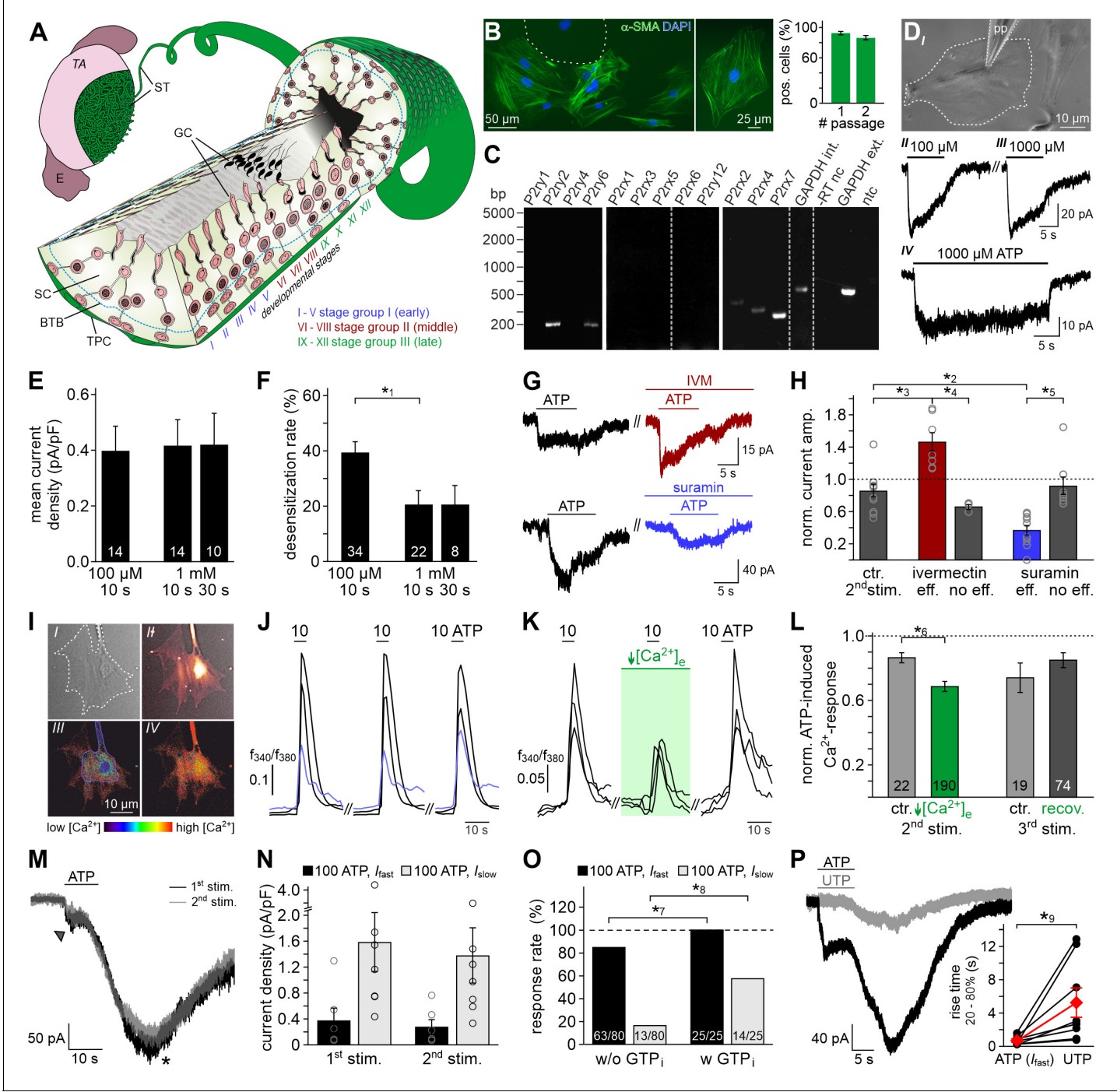

**Figure 1.** ATP is a potent TPC stimulus. (**A**) Schematic sketch of the mouse seminiferous tubule (ST) highlighting 12 stages (I – XII) of the spermatogenic cycle (**Hess and De Franca, 2008**; **Russell, 1990**), which are arranged in consecutive order along the length of the tubule. A single layer of flat testicular peritubular cells (TPC; green) lines the tubular wall. Sertoli cells (SC) span the tubule from the basal lamina to the lumen. Developing germ cells (GC) are distributed between Sertoli cells. Spermatogonia are located near the basal membrane. Prophase spermatocytes move across the blood-testis barrier (BTB) to the adluminal compartment where they complete meiosis. The resulting haploid spherical cells (round spermatids) differentiate into elongated spermatids and, eventually, into highly condensed and compartmentalized spermatozoa (spermiogenesis). These mature, yet still immotile germ cells are then released into the lumen (spermiation). E, epididymis; TA, *tunica albuginea*; inspired by **Hess and De Franca, 2008**. (**B**) Immunostaining against α-smooth muscle actin (α-SMA, green) marks TPCs in vitro (**Tung and Fritz, 1989**). Cell count is determined by nuclear staining (DAPI, blue). Cultures retain high TPC purity for at least two passages (92 ± 2%, n = 1102 (#1); 86 ± 3%, n = 542 (#2)). Dashed line delimits one of the few α-SMA-negative cells. (**C**) RT-PCR profiling of purinoceptor isoforms in TPC cultures reveals P2rx2, P2rx4, P2rx7, P2ry2, and P2ry6 transcripts. Dashed gray vertical lines indicate cuts in a given gel. (**D–H**) ATP exposure triggers TPC transmembrane currents. (**D$_I$**) Phase-contrast micrograph

*Figure 1 continued on next page*

*Figure 1 continued*

depicting a TPC (dashed line) targeted by a patch pipette (pp). ($D_{II}$-$D_{IV}$) Original whole-cell recordings illustrate representative currents in response to ATP stimulation (100 µM ($D_{II}$) vs. 1000 µM ($D_{III}$) and 10 s ($D_{III}$) vs. 30 s ($D_{IV}$), respectively). $V_{hold}$ = −80 mV. (E, F) Quantification (bar charts; mean ± SEM; n as indicated) reveals saturation of peak current density at ≤100 µM ATP (E) and modest desensitization at a concentration-dependent rate (F). (G) Whole-cell voltage-clamp recordings show ATP-induced currents (100 µM; 10 s) that are potentiated by ivermectin (3 µM) and partially inhibited by suramin (100 µM), respectively (≥60 s preincubation). $V_{hold}$ = −80 mV. (H) Quantification (bar charts; mean ± SEM; data normalized to initial control response) demonstrates dichotomy in drug sensitivity. Treatment was categorized as effective (eff) if current amplitudes deviate by ± SD from average control recordings (85 ± 24%, 2nd ATP stimulation). Note that each drug proved ineffective (no eff) in some cells. Gray circles depict data from individual cells. (I–L) ATP-dependent $Ca^{2+}$ mobilization in cultured TPCs. $Ca^{2+}$ transients in response to repetitive stimulation (10 µM, 10 s) are monitored by ratiometric (fura-2) fluorescence imaging. (I) Phase contrast (*I*) and merged fluorescence ($f_{380}$; *II*) images of a TPC in vitro. Bottom pseudocolor frames (rainbow 256 color map) illustrate relative cytosolic $Ca^{2+}$ concentration ($[Ca^{2+}]_c$) before (*III*) and during (*IV*) ATP stimulation. (J, K) Representative original traces from time-lapse fluorescence ratio ($f_{340}/f_{380}$) recordings depict repetitive $[Ca^{2+}]_c$ elevations upon ATP exposure under control conditions ((J) blue traces correspond to the TPC in (I)) and during reduced extracellular $Ca^{2+}$ concentration ((K) $[Ca^{2+}]_e$ = 100 nM; 60 s preincubation). (L) Bar chart depicting $Ca^{2+}$ signal amplitudes (mean ± SEM; n as indicated) – normalized to the initial ATP response – under control conditions (gray) vs. low $[Ca^{2+}]_e$ (green). Asterisks denote statistically significant differences (*[1]p=0.001; *[2]p=0.002; *[3]p=5.5e$^{-5}$; *[4]p=0.0006; *[5]p=0.02; *[6]p=0.02; Student *t*-test (F, L), one-way ANOVA (H)). (M) Representative whole-cell voltage-clamp recordings ($V_{hold}$ = −80 mV) of ATP-induced inward currents in cultured mouse TPCs. Two components – a fast relatively small current (arrow head) and a delayed lasting current (asterisk) – are triggered repeatedly by successive ATP exposure (100 µM; 90 s inter-stimulus interval). Notably, we never observed a delayed slow current without a fast response. (N) Bar chart quantifying peak densities (mean ± SEM, circles show individual values) of the fast ($I_{fast}$; black) and the delayed ($I_{slow}$; gray) ATP-induced current components (1st stimulation: $I_{fast}$ 0.37 ± 0.2 pA/pF; $I_{slow}$ 1.58 ± 0.5 pA/pF; 2nd stimulation $I_{fast}$ 0.27 ± 0.1 pA/pF; $I_{slow}$ 1.37 ± 0.4 pA/pF). (O) Bar graph illustrating the frequency of $I_{fast}$ (black) and $I_{slow}$ (gray) occurrence upon ATP (100 µM) stimulation in absence (w/o) and presence (w) of GTP (500 µM) in the pipette solution, respectively. Asterisks denote statistically significant differences (*[7]p=0.008, *[8]p=0.0003; Fisher's exact test); n as indicated in bars. (P) Representative whole-cell voltage-clamp recordings ($V_{hold}$ = −80 mV) of inward currents induced by ATP (100 µM) and UTP (100 µM), respectively. Whenever ATP triggers both $I_{fast}$ and $I_{slow}$ (left), $I_{slow}$ is also induced by UTP (right). UTP-dependent currents develop significantly slower than ATP-evoked $I_{fast}$ (inset; *[9]p=0.03; paired *t*-test).

The online version of this article includes the following figure supplement(s) for figure 1:

**Figure supplement 1.** Mouse TPCs in primary culture.

**Figure supplement 2.** ATP/UTP-dependent $I_{slow}$ in cultured mouse TPCs is largely carried by Cl$^-$.

While bulk movement of luminal content has been anecdotally reported (*Cross, 1958*; *Setchell et al., 1978*; *Worley et al., 1985*), no quantitative data on sperm transport within the semi-niferous tubules is available. Early in vitro observations of apparent minute undulating motions of seminiferous tubule segments (*Roosen-Runge, 1951*; *Suvanto and Kormano, 1970*) suggested that smooth muscle-like TPCs (*Clermont, 1958*; *Ross, 1967*) could mediate contractile tubule move-ments. This concept has gained widespread support from several, mostly indirect, in vitro studies (*Ailenberg et al., 1990*; *Filippini et al., 1993*; *Miyake et al., 1986*; *Tripiciano et al., 1996*). How-ever, quantitative direct (i.e. live cell) measurements of seminiferous tubule contractions are rare and controversial (*Ellis et al., 1978*; *Harris and Nicholson, 1998*; *Losinno et al., 2012*; *Worley and Leendertz, 1988*). Moreover, mechanistic in vivo evidence is lacking. Here, we demonstrate that, by acting on ionotropic and metabotropic P2 receptors, extracellular ATP activates TPC contractions that trigger directional sperm movement within the mouse seminiferous tubules both in vitro and in vivo.

## Results

### ATP is a potent TPC stimulus

Accumulating data suggests that purinergic signaling constitutes a critical component of testicular paracrine communication (*Fleck et al., 2016*; *Foresta et al., 1995*; *Gelain et al., 2003*; *Poletto Chaves et al., 2006*; *Veitinger et al., 2011*; *Walenta et al., 2018*), with Sertoli cells acting as a primary source of ATP secretion (*Gelain et al., 2005*). Therefore, we asked if mouse TPCs are sensitive to extracellular ATP. Primary TPC cultures retain high purity for ≥14 days in vitro (*Figure 1B*, and *Figure 1—figure supplement 1A&B*) and cells express transcripts for several iono-tropic (P2X2, P2X4, P2X7) and metabotropic (P2Y2, P2Y6) purinoceptors (*Figure 1C*). The specific biophysical and pharmacological profile of ATP-dependent transmembrane currents (*Figure 1D–H*) strongly suggests functional expression of P2X2 and/or P2X4, but not P2X7 receptors. As reported for both P2X2 and P2X4 (*North, 2002*), TPC currents are saturated at ≤100 µM ATP (*Figure 1D&E*),

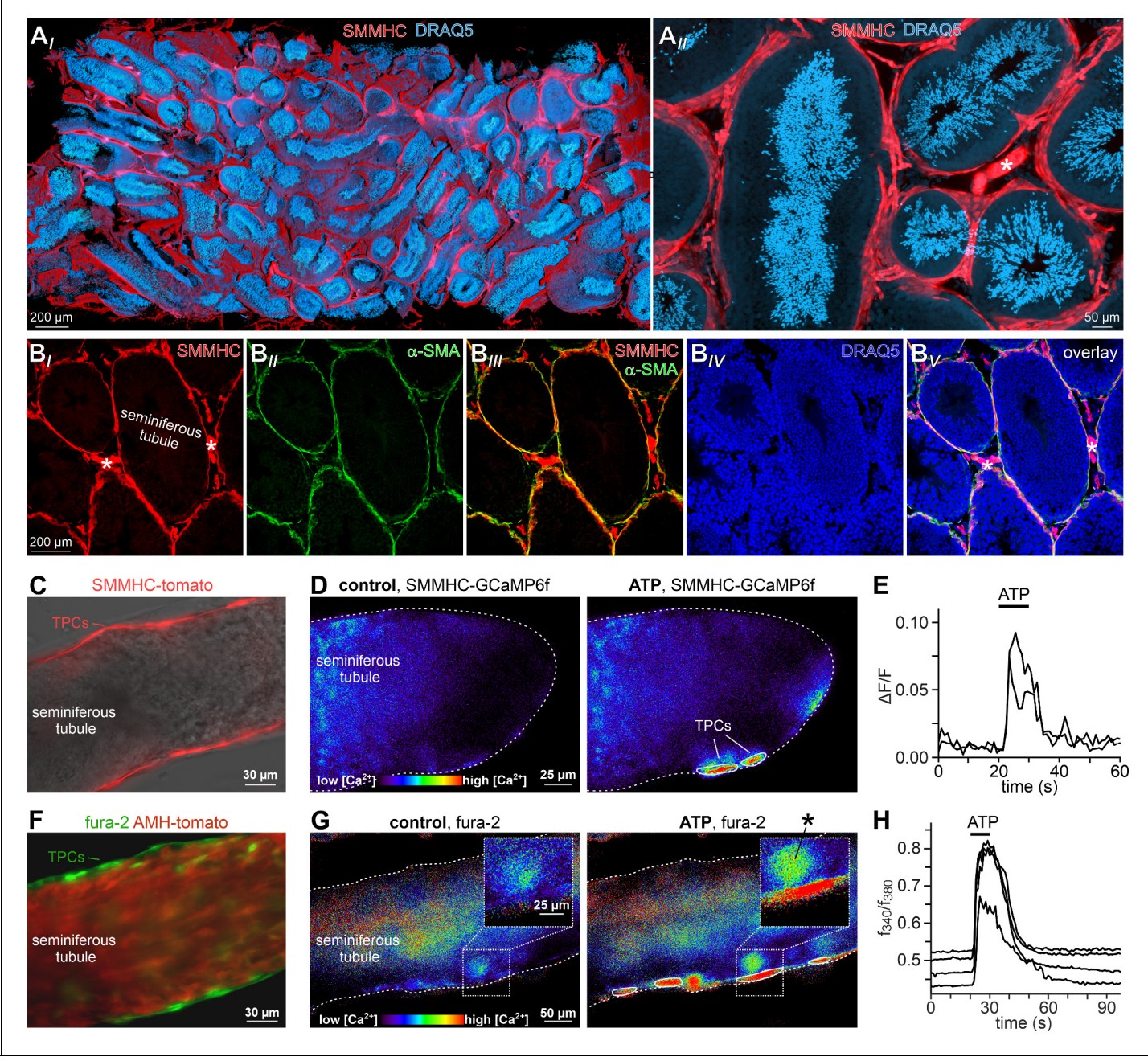

**Figure 2.** ATP triggers TPC Ca$^{2+}$ signals in acute seminiferous tubule sections. (**A**) 3D reconstruction of an intact 6 × 3 × 1.5 mm testis sample from a SMMHC-CreER$^{T2}$ x Ai14D mouse after tissue clearing (CLARITY ***Chung and Deisseroth, 2013***) reveals tdTomato expression (red) restricted to TPCs and vascular endothelial cells (asterisk in (**A**$_{II}$)). Nuclear staining (DRAQ5; blue) is most prominent in post-meiotic germ cells because of their high degree of DNA condensation. (**B**) SMMHC-CreER$^{T2}$-expressing cells in the tubule wall are TPCs. In testis cryosections from adult SMMHC-CreER$^{T2}$ x Ai14D mice, Cre-driven tdTomato signals (**B**$_I$) and α-SMA immunostaining (**B**$_{II}$) colocalize at tubular margins (**B**$_{III}$). Nuclei are stained with DRAQ5 (blue (**B**$_{IV}$)). Note that endothelial vasculature in interstitial regions (asterisks) is α-SMA-negative (merged image (**B**$_V$)). (**C–H**) Both TPC-specific expression of a genetically encoded Ca$^{2+}$ indicator (GCaMP6f) and bulk loading with a synthetic Ca$^{2+}$ sensor (fura-2) allow for TPC-selective live cell Ca$^{2+}$ imaging in acute seminiferous tubule sections. (**C**) Merged fluorescence and reflected light micrographs show the location of SMMHC-expressing TPCs (red) within the wall of an intact tubule. (**D–E**) Cre-dependent GCaMP6f expression in SMMHC-CreER$^{T2}$ x Ai95D mice reveals Ca$^{2+}$ transients in TPCs in response to ATP. Representative fluorescence images ((**D**) rainbow 256 color map) before and during ATP exposure (100 µM; 10 s), and corresponding traces (**E**) showing changes in GCaMP6f intensity (ΔF/F) over time. Traces from ROIs outlined in (**D**) (white solid lines). (**F**) Merged fluorescence image of an acute seminiferous tubule section from an AMH-Cre x Ai14D mouse after bulk loading with fura-2/AM (green). Anti-Müllerian hormone (AMH) dependent expression of tdTomato (red) specifically labels Sertoli cells that build the seminiferous epithelium. Note the narrow green band of marginal TPCs that are preferentially labeled by the Ca$^{2+}$-sensitive dye. (**G–H**) Ratiometric Ca$^{2+}$ imaging in fura-2-loaded tubules enables semi-quantitative live-cell

*Figure 2 continued on next page*

*Figure 2 continued*

monitoring of TPC activity. Representative fluorescence images ((**G**) rainbow 256 color map) before and during ATP exposure (100 μM; 10 s). Corresponding traces (**H**) show the fluorescence intensity ratio ($f_{340}/f_{380}$) from four ROIs (in (**G**); white solid lines) over time. Inset (**G**) shows a putative TPC and an adjacent putative spermatogonium (asterisk) at higher magnification.

whereas P2X7 receptors display strongly reduced ATP sensitivity (*Donnelly-Roberts et al., 2009*). Moreover, currents recorded from TPCs showed modest but persistent desensitization (*Figure 1D&F*), which is similarly observed for recombinant P2X2 and P2X4, but not P2X7 receptors (*Coddou et al., 2011*). TPCs also displayed reduced BzATP sensitivity (data not shown), which is a potent activator of P2X7 receptors (*Donnelly-Roberts et al., 2009*). Ivermectin (*Figure 1G&H*), an agent selectively potentiating P2X4 receptor currents (*Khakh et al., 1999*; *Silberberg et al., 2007*), increased ATP-induced currents in a subpopulation of TPCs (n = 7/12), whereas suramin (*Figure 1G&H*), a drug inhibiting P2X2, but not P2X4 receptors (*Evans et al., 1995*), inhibited a TPC subset (n = 10/18).

Notably, live-cell ratiometric $Ca^{2+}$ imaging in cultured TPCs revealed robust and repetitive cytosolic $Ca^{2+}$ transients upon ATP exposure (*Figure 1I&J*). We next reduced the extracellular $Ca^{2+}$ concentration ($[Ca^{2+}]_e$) to 100 nM, a concentration approximately equimolar to cytosolic levels, by adding an appropriate chelator/ion concentration ratio (1 mM EGTA/0.5 mM $CaCl_2$). This treatment, which drastically diminishes the driving force for $Ca^{2+}$ influx, did substantially reduce, but not abolish ATP response amplitudes (*Figure 1K&L*). The selective P2Y receptor agonist UTP (*Alexander et al., 2019b*) also triggered $Ca^{2+}$ signals (data not shown), indicating a role for G protein-dependent $Ca^{2+}$ release from internal storage organelles (*Müller et al., 2020*). Notably, ~46% of all ATP-sensitive TPCs additionally displayed a delayed, but long-lasting inward current that gradually developed over tens of seconds after ATP stimulation ended (*Figure 1M&N*). We hypothesized that this slower current could result from P2Y receptor-/G protein-dependent $Ca^{2+}$ release, likely mediated by the P2Y2 isoform since P2Y6 receptors lack substantial ATP sensitivity (*Alexander et al., 2019a*; *Jacobson et al., 2015*). Indeed, occurrence of the delayed current depends on presence of intracellular GTP (*Figure 1O*). Moreover, selective recruitment of G protein-coupled P2Y receptors with UTP (*Figure 1P*) exclusively triggered such slowly developing currents. Largely carried by $Cl^-$ (*Figure 1—figure supplement 2*), this current likely results from P2Y receptor-mediated phosphoinositide turnover, $Ca^{2+}$ release, and activation of $Ca^{2+}$-gated $Cl^-$ channels. Together, these data suggest that mouse TPCs functionally express both ionotropic and metabotropic purinoceptors.

Next, we asked if TPCs also exhibit ATP sensitivity in their physiological setting. Therefore, we examined purinergic $Ca^{2+}$ signals from mouse TPCs in acute seminiferous tubule sections (*Fleck et al., 2016*). In parallel approaches, we employed two different fluorescent $Ca^{2+}$ reporters, a synthetic ratiometric $Ca^{2+}$ sensor (fura-2) as well as a genetically encoded $Ca^{2+}$ indicator (GCaMP6f). The dual excitation ratiometric indicator fura-2 allows semi-quantitative $Ca^{2+}$ measurements (*Bootman et al., 2013*), but lacks cell type specificity as tubules are bulk-loaded with a membrane-permeable acetoxymethyl ester conjugate. By contrast, conditional gene targeting via the Cre/Lox system (*Smith, 2011*) allows TPC-specific expression of the single-wavelength indicator GCaMP6f. First, we confirmed inducible TPC-targeted testicular expression of fluorescent reporter proteins in SMMHC-CreER^T2 x Ai14D mice (*Figure 2A–C*, *Video 1*). Tamoxifen-induced transgenic expression of CreER^T2 under control of the mouse smooth muscle myosin, heavy polypeptide 11 (a.k.a. SMMHC) promoter drives *Cre*-mediated recombination of *loxP*-flanked reporters (tdTomato (Ai14D mice) or GCaMP6f (Ai95D))

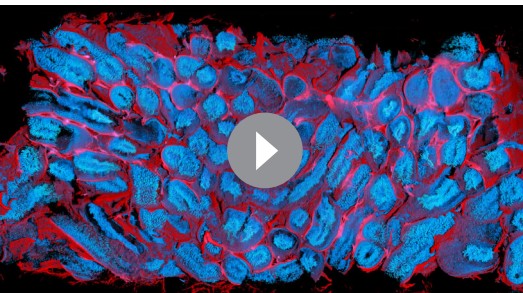

**Video 1.** SMMHC-CreER^T2 mice allow inducible TPC-specific expression of genetically encoded fluorescent reporter proteins. After tamoxifen injections, SMMHC-CreER^T2 x Ai14D male offspring express tdTomato (red) in both TPCs and vascular smooth muscle cells. Video shows the 3D reconstruction of an intact and cleared (CLARITY, *Chung and Deisseroth, 2013*) 6 × 3 × 1.5 mm testis sample with nuclei labeled by DRAQ5 (blue). https://elifesciences.org/articles/62885#video1

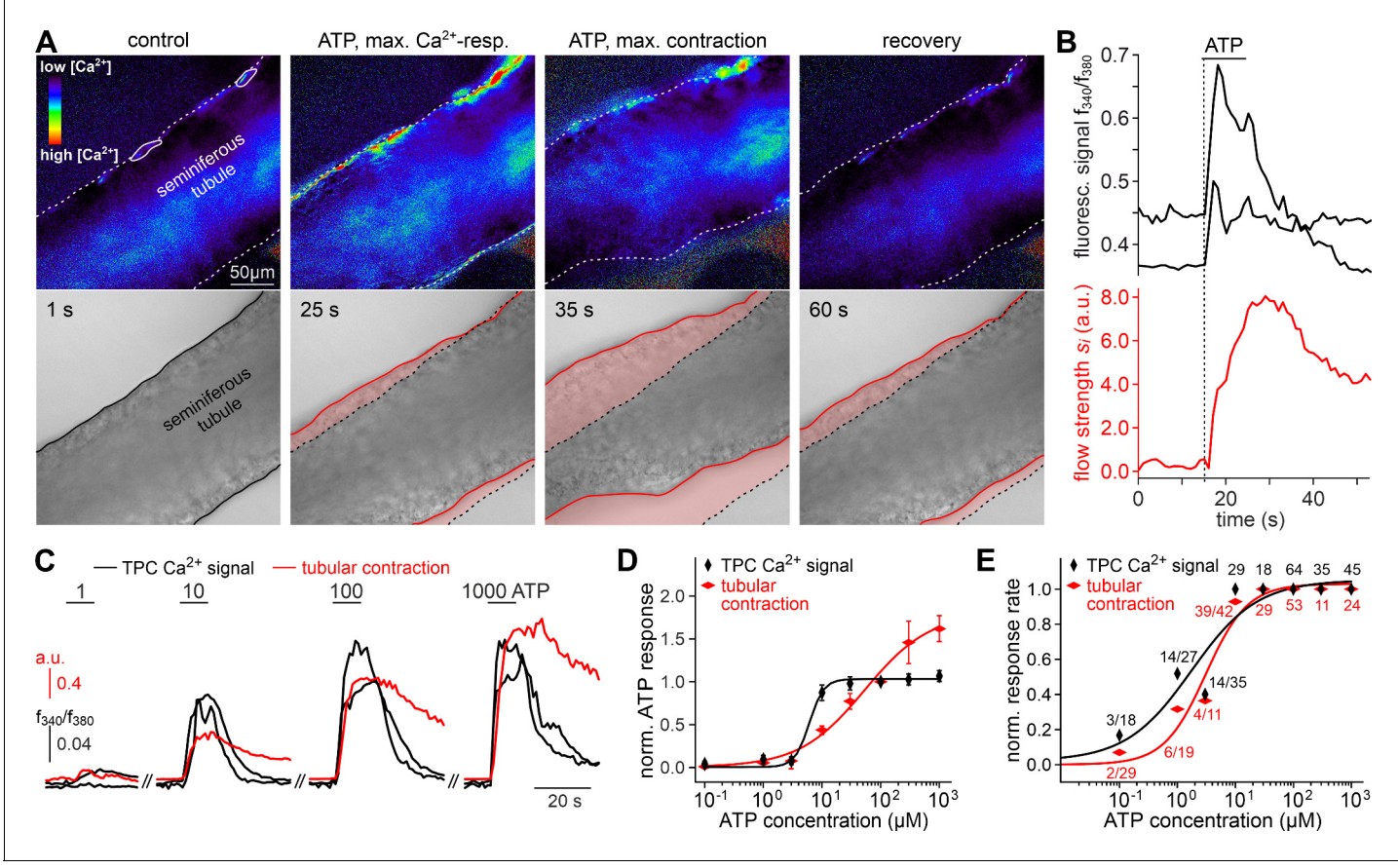

**Figure 3.** ATP triggers seminiferous tubule contractions. (**A**) Quasi-simultaneous imaging of $[Ca^{2+}]_c$-dependent fluorescence (top; $f_{340}/f_{380}$; rainbow 256 color map) and tubular position (bottom; reflected light microscopy). Focus adjusted to provide sharp images of the seminiferous tubule's edges. Individual frames correspond to the time points indicated, i.e. before, during, and after ATP exposure (see (**B**)). Dashed white lines (top) and corresponding solid black/red lines (bottom) depict the outline of the tubule in each image, respectively. Dotted black lines (bottom) show the contour at t = 1 s for comparison. Pink shades (bottom) accentuate areas that moved. (**B**) Fluorescence ratio (top; black traces correspond to regions-of-interest delimited by solid white lines in (**A**)) and integrated flow strength $s_i$ – a measure of strength and direction of pixel displacement (bottom; red trace) – over time. ATP (100 µM) stimulation as indicated (horizontal bar). With the t = 0 s image as reference, flow strength $s_i$ is calculated by custom code as the average whole tubule pixel shift vector (methods). Dashed vertical line marks the $Ca^{2+}$ signal onset. (**C–E**) $Ca^{2+}$ responses and tubular movement are dose-dependent. (**C**) Original traces depict $[Ca^{2+}]_c$ (black) and tubule movement (red) from a representative experiment. Data calculated as in (**B**). Brief (10 s) stimulations with increasing ATP concentrations (1–1000 µM) trigger dose-dependent $Ca^{2+}$ transients and corresponding contractions. (**D, E**) Data quantification. Dose-response curves illustrate peak signals (**D**) and the percentage of responding putative TPCs (black)/tubules (red) (**E**). Data are normalized to responses to 100 µM ATP (n as indicated in (**E**)).

in smooth muscle cells and TPCs (*Wirth et al., 2008*). Second, TPC-specific GCaMP6f expression in SMMHC-CreER[T2] x Ai95D mice revealed robust $Ca^{2+}$ transients in cells of the tubular wall upon ATP exposure (*Figure 2D&E*). Third, fura-2/AM loading of acute seminiferous tubule sections preferentially labeled the outermost cell layer (*Figure 2F*), allowing semi-quantitative in situ imaging of ATP-dependent $Ca^{2+}$ signals in mouse TPCs (*Figure 2G&H*). So far, our results thus demonstrate that challenging TPCs with extracellular ATP triggers robust $Ca^{2+}$ signals both in vitro and in situ.

## ATP triggers seminiferous tubule contractions

We hypothesized that ATP-induced $Ca^{2+}$ signals in TPCs could mediate contractile motion of the seminiferous tubule. To address this, we established a fast, quasi-simultaneous image acquisition method that enables parallel recording of both peritubular $Ca^{2+}$ responses and seminiferous tubule movement (methods). Brief ATP exposure resulted in a peripheral band of $Ca^{2+}$ activity at the edge of the tubule. Such signals usually coincided with a pronounced contractile motion of the seminiferous tubule (*Figure 3A*, *Video 2*). When movement is quantified as the time-lapse image flow field

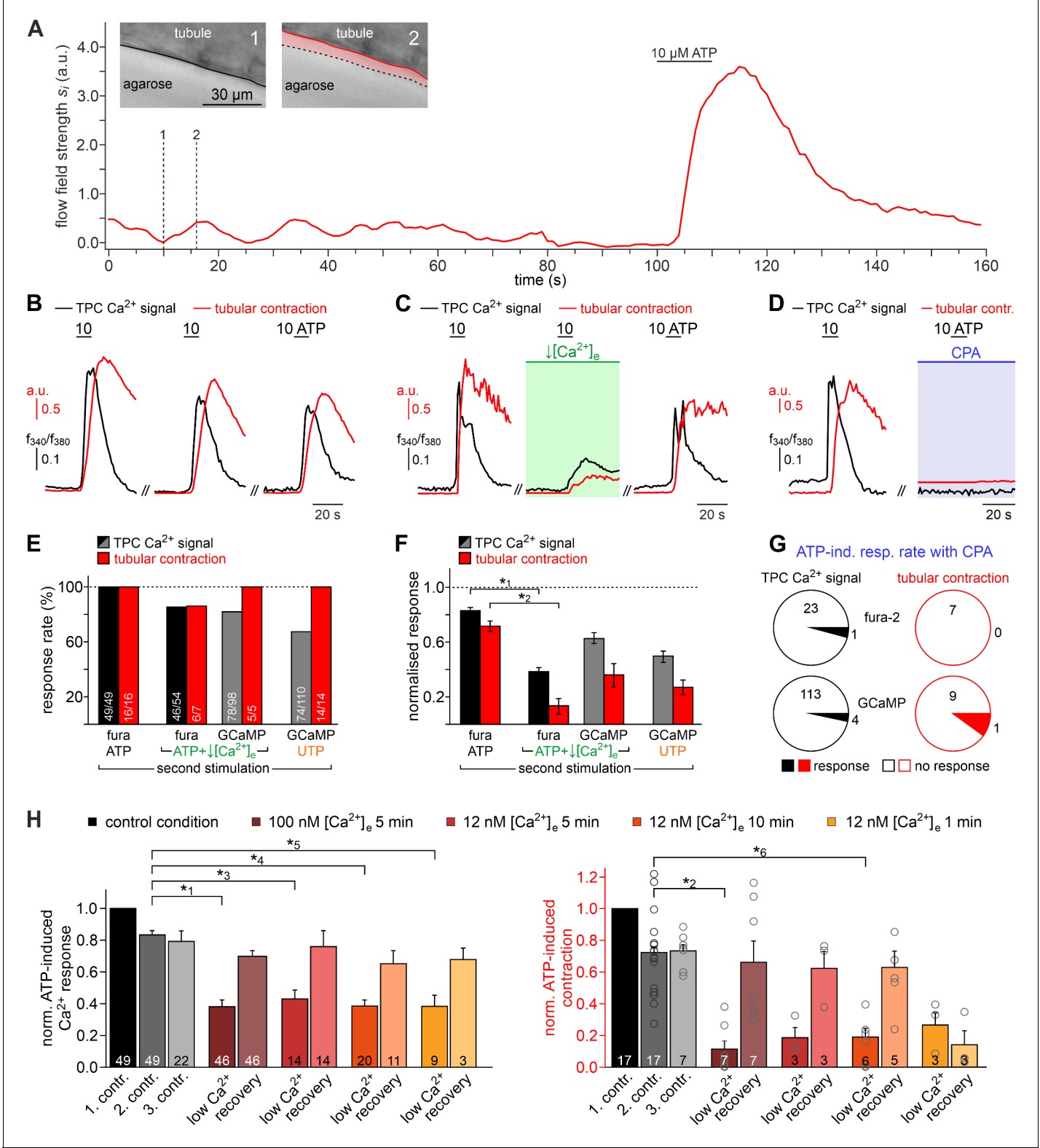

**Figure 4.** Both intra- and extracellular Ca²⁺ sources contribute to ATP-dependent TPC contractions. (**A**) In situ imaging identifies spontaneous low-amplitude 'vibratory' movements in acute seminiferous tubule slices. Representative trace illustrating flow field strength analysis of tubular motion under control conditions and upon ATP exposure (10 μM; 10 s). Note that spontaneous indentations share small amplitudes and are restricted to the tubule edge (inset). Black/red lines (inset) depict the outline of the tubule in each image, respectively. Dotted black lines show the contour at t = 1 for comparison. Pink shades accentuate areas that moved. (**B**) Repeated purinergic stimulation triggers robust Ca²⁺ elevations and concurring seminiferous

*Figure 4 continued*

tubule contractions with only minor response adaptation. Traces depict changes in fura-2 intensity ratio ($f_{340}/f_{380}$; black) or tubular movement (red) upon brief ATP exposure (10 s; 10 µM; 5 min intervals) under control conditions. (**C**) Reducing [$Ca^{2+}$]$_e$ (100 nM; 5 min incubation) strongly diminishes responses to ATP (10 s; 10 µM). (**D**) Depletion of internal $Ca^{2+}$ stores (CPA *Seidler et al., 1989*; 90 µM; 18.8 ± 9.3 min incubation) essentially abolishes both $Ca^{2+}$ signals and tubule contractions. (**E–G**) Quantification of data exemplified in (**B–D**). (**E**) Bar chart depicting ATP sensitivity (response rate; %), independent of signal strength. Occurrence of $Ca^{2+}$ elevations (black) and tubule contractions (red) are plotted for different experimental conditions [i.e. stimulation with ATP or UTP, regular or reduced [$Ca^{2+}$]$_e$ (1 mM or 100 nM, respectively), and $Ca^{2+}$ indicator (fura-2 or GCaMP6f, respectively)]. Numbers of experiments are indicated in each bar. (**F**) Signal amplitudes ($Ca^{2+}$, black; contractions, red) of responding TPCs/tubules, quantified as a function of stimulus, treatment, and $Ca^{2+}$ sensor. Data (mean ± SEM) are normalized to the respective initial responses to ATP (10 µM) under control conditions (dotted horizontal line; see first stimulations in **B** and **C**). Experimental conditions and numbers of experiments as in (**E**). Asterisks denote statistical significance (*[1]p=2.2e$^{-19}$ and *[2]p=5.6e$^{-8}$; *t*-test; note: tests only performed when n > 5 and only one variable was changed). (**G**) Pie charts illustrating the profoundly reduced ATP sensitivity of TPCs/tubules after depletion of $Ca^{2+}$ storage organelles (CPA; 90 µM). Numbers within pies correspond the total count of cells/tubules that responded to ATP before treatment. (**H**) Effects of lowering [$Ca^{2+}$]$_e$ are comparable over both incubation periods and concentrations in the nanomolar range. Significantly reduced, though not abolished TPC $Ca^{2+}$ signals (left) and seminiferous tubule contractions (right) are observed in presence of both 100 nM and 12 nM [$Ca^{2+}$]$_e$ as well as for variable incubation periods lasting between 1 and 10 min, respectively. Asterisks denote statistical significance (*[3]p=2.8e$^{-10}$, *[4]p=7.8e$^{-14}$, *[5]p=3.8e$^{-9}$, *[6]p=1.0e$^{-6}$; one-way ANOVA with *post-hoc* Tukey HSD test; note: tests only performed when n ≥ 5).

The online version of this article includes the following figure supplement(s) for figure 4:

**Figure supplement 1.** Purinergic stimulation mediates contractions in cultured human TPCs.

strength (methods) tubular contraction follows the $Ca^{2+}$ signal onset with minimal delay, outlasts the $Ca^{2+}$ signal peak, and recovers slowly (*Figure 3B*). Both $Ca^{2+}$ responses and tubular movement are dose-dependent and share an ATP threshold concentration of approximately 1 µM (*Figure 3C–E*, *Video 3*). Contractile smooth muscle plasticity (*Tuna et al., 2012*) likely underlies the apparent difference in signal saturation (*Figure 3D*). Notably, in some tubules, we observed spontaneous low-

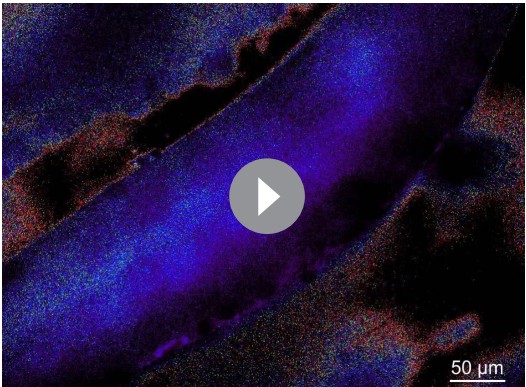

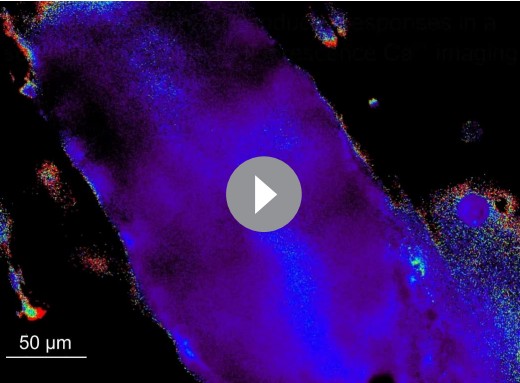

**Video 2.** Quasi-simultaneous recording of peritubular $Ca^{2+}$ signals and seminiferous tubule movement. A representative seminiferous tubule section (250 µm) is stimulated with ATP (100 µM, 10 s). After fura-2 bulk loading, ratiometric fluorescence imaging ($f_{340}/f_{380}$) reveals relative changes in $Ca^{2+}$ concentration (rainbow color map; blue, low $Ca^{2+}$; red, high $Ca^{2+}$) in a peripheral band of putative TPCs at the tubule's edge. Since each image acquisition cycle (1 Hz) captures two fluorescence (Exλ$_{340}$; Exλ$_{380}$) and one reflective light image (brightfield), time-lapse recordings allow parallel physiological phenotyping of both seminiferous tubule $Ca^{2+}$ responses and movement (shown sequentially for clarity).
https://elifesciences.org/articles/62885#video2

**Video 3.** Both ATP-induced seminiferous tubule $Ca^{2+}$ responses and contractions are dose-dependent. A representative seminiferous tubule section (250 µm; fura-2 bulk loading) is stimulated with increasing ATP concentrations (1–1000 µM, 10 s). Ratiometric fluorescence imaging ($f_{340}/f_{380}$) reveals relative changes in $Ca^{2+}$ concentration (rainbow color map; blue, low $Ca^{2+}$; red, high $Ca^{2+}$) in putative TPCs. Quasi-simultaneous time-lapse recording of fluorescence (Exλ$_{340}$; Exλ$_{380}$) and brightfield (reflective light) images illustrates that both seminiferous tubule $Ca^{2+}$ signals and contractions (shown sequentially for clarity) are dose-dependent and share an ATP threshold concentration of approximately 1 µM.
https://elifesciences.org/articles/62885#video3

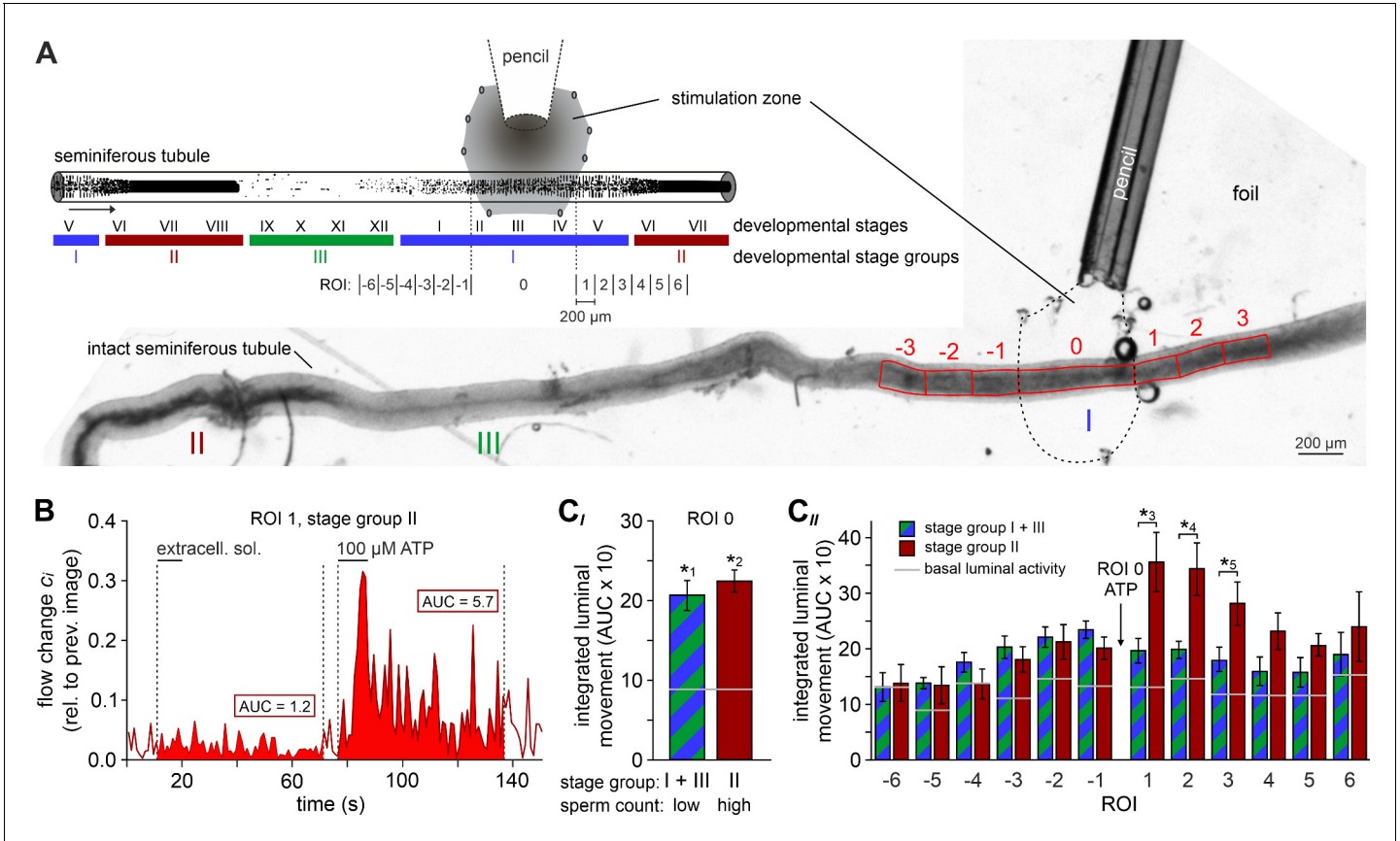

**Figure 5.** ATP drives directional luminal transport. (A) Schematic drawing (top) and original low-magnification image (bottom) of the experimental setup. Intact seminiferous tubules are placed on transparent foil in a custom-built macroscopic imaging chamber. The tubule is kept stationary by gentle suction through tiny holes punched in the foil and vacuum underneath. As previously suggested (**Hess and De Franca, 2008**), tubules are coarsely categorized into three stages (I–III; color code) according to luminal sperm content. Precise mapping of stimulated regions is feasible by positioning both tubule and perfusion pencil within an area delimited by several holes that outline a stimulation zone (methods). The tubule region directly exposed to ATP is designated as ROI 0, with adjacent equidistant sections numbered consecutively (up to ROI ±6; schematic). In the original image shown, only ROIs ± 3 are outlined for clarity. (B) Analysis of luminal content movement by calculation of flow change $c_i$ relative to each previous image (methods) within a representative luminal ROI. Motion is quantified by measuring the area under curve (AUC; solid red) within 60 s after stimulation onset. Note that mechanical control stimulation (extracellular solution) does not affect basal luminal motion. (C) Bar charts depicting luminal content movement (means ± SEM) upon ATP stimulation (100 μM; 10 s) in either directly exposed regions ($C_I$; n = 17) or adjacent areas ($C_{II}$; n = 3–17). Green/blue (groups I and III) and red (group II) bars depict stages with a low vs. a high luminal sperm count, respectively. Horizontal gray lines mark the average basal luminal motion prior to stimulation. ATP induces significantly increased content movement in directly stimulated areas (ROI 0) independent of luminal sperm count/stage group ($C_I$). Note that in adjacent regions ($C_{II}$) unidirectional movement occurs exclusively in tubule sections with high luminal sperm density. Asterisks denote statistically significant differences (*[1]p=8.7e$^{-5}$; *[2]p=6.7e$^{-7}$; *[3]p=0.005; *[4]p=0.002; *[5]p=0.03; unpaired two-tailed *t*-test).

amplitude 'vibratory' movements and local indentations (**Figure 4A**), reminiscent of the relatively high frequency rippling previously described (**Ellis et al., 1981**; **Worley et al., 1985**).

We next investigated the Ca$^{2+}$ signaling mechanism(s) underlying ATP-dependent TPC contractions. First, we asked whether influx of external Ca$^{2+}$ is involved in TPC force generation. Similar to in vitro observations (**Figure 1K&L**), diminishing or even reversing the driving force for transmembrane Ca$^{2+}$ flux by reducing [Ca$^{2+}$]$_e$ to 100 nM or 12 nM, respectively, for variable durations, significantly decreased both TPC Ca$^{2+}$ signals and tubular contractions (**Figure 4B–H**). While, upon [Ca$^{2+}$]$_e$ reduction, ATP-dependent responses (both Ca$^{2+}$ signals and contractions) were still detected in the vast majority of cells/experiments (**Figure 4E**), response strength was strongly diminished (**Figure 4F**). These effects were independent of both the extent (12 nM or 100 nM) and the duration (1–10 min) of [Ca$^{2+}$]$_e$ reduction and were fully reversible (**Figure 4H**). Second, we examined a potential role of ATP-induced Ca$^{2+}$ release from internal storage organelles. Ca$^{2+}$ depletion of the

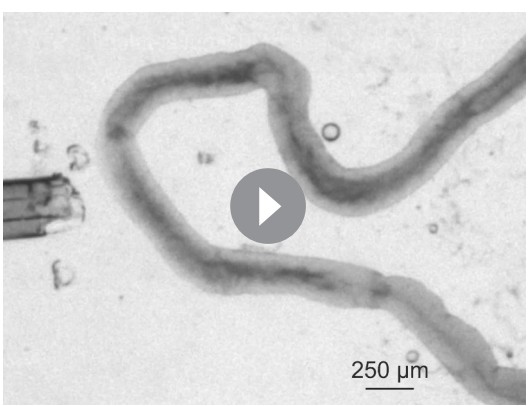

**Video 4.** ATP stimulation triggers movement of luminal content in intact seminiferous tubules. Brightfield time-lapse recording of an intact isolated seminiferous tubule (field of view shows cycle stages II and III) challenged by brief focal ATP perfusion (100 µM, 10 s). The spatial extent of the stimulation zone had been defined by prior perfusion with a dye solution (fast green).
https://elifesciences.org/articles/62885#video4

sarcoplasmic reticulum via pharmacological inhibition of the sarco/endoplasmic reticulum $Ca^{2+}$-ATPase by cyclopiazonic-acid (CPA) essentially abolished both ATP-dependent $Ca^{2+}$ signals and contractions (*Figure 4D*), with very few cells/tubules retaining some residual ATP sensitivity during CPA treatment (*Figure 4G*). Importantly, all results from ratiometric fura-2 imaging were qualitatively indistinguishable from those obtained with genetically targeted GCaMP6f (*Figure 4E–G*), showing that both approaches to TPC $Ca^{2+}$ measurement provide comparable results. Third, given the pronounced effect of pharmacological store depletion, we aimed to quantify the specific contribution of metabotropic purinoceptors to the overall ATP-mediated effect. The P2Y receptor-selective agonist UTP (*Coddou et al., 2011*) evoked both TPC $Ca^{2+}$ signals and tubular contractions (*Figure 4E&F*). However, under control $[Ca^{2+}]_e$ conditions, UTP-evoked responses were substantially reduced compared to control ATP stimulations (*Figure 4F*). Notably, these UTP responses were statistically indistinguishable from the diminished ATP-dependent signals we observed under low $[Ca^{2+}]_e$ conditions (*Figure 4F*).

Together, these data strongly suggest that (i) extracellular ATP acts as a potent TPC stimulus that triggers seminiferous tubule contractions in situ, that (ii) P2X and P2Y receptors act in concert to mediate TPC responses to ATP exposure, that (iii), while P2X receptor-dependent external $Ca^{2+}$ influx apparently boosts responses to ATP, P2Y receptor-mediated $Ca^{2+}$ mobilization from the sarcoplasmic reticulum is necessary to evoke TPC responses, and consequently – since store depletion essentially abolishes ATP-dependent signals – that (iv) influx of external $Ca^{2+}$ via ionotropic P2X receptors is not sufficient to drive TPC signals and evoke contractions. Notably, our general finding of ATP-induced mouse TPC contractions is likely transferable to human peritubular cells. When primary human TPC cultures (*Walenta et al., 2018*) were exposed to extracellular ATP, morphological changes were observed within seconds-to-minutes (*Figure 4—figure supplement 1A&B*). Moreover, embedding cells in collagen gel lattices revealed considerable contractile force in response to ATP (*Figure 4—figure supplement 1C&D*).

## ATP drives directional luminal transport

We hypothesized that ATP-induced tubular contractions could impact the transport of luminal fluid and spermatozoa. To test this, we custom-built a whole-mount macroscopic imaging platform, designed to allow both widefield and fluorescence time-lapse imaging of intact seminiferous tubules (*Figure 5A*). In addition, this setup enables visual categorization of the spermatogenic cycle into three distinct stage groups following published protocols (*Hess and De Franca, 2008*) and allows precisely timed focal perfusion (methods). First, we asked if brief focal purinergic stimulation triggers seminiferous tubule contractions and, consequently, luminal content movement. Flow field change analysis reveals some basal luminal motion independent of mechanical stimulation (*Figure 5B*). However, ATP exposure triggered a strong increase in luminal flow that outlasted the presence of ATP for several tens of seconds (*Figure 5B*, *Video 4*). Second, we analyzed if luminal movement depends on the tubule's cycle stage and, consequently, luminal sperm count. When we analyzed ATP-induced movement in directly stimulated luminal regions (each designated as region-of-interest (ROI) 0) and compared stage groups with a high (group II) vs. a relatively low (groups I and III) amount of luminal sperm, we observed no difference in stimulation-dependent motion (*Figure 5C_I*). Thus, direct ATP exposure triggers tubular contractions independent of cycle stage and luminal sperm count. Third, we investigated if luminal movement is restricted to the area of stimulation or, by contrast, if fluid

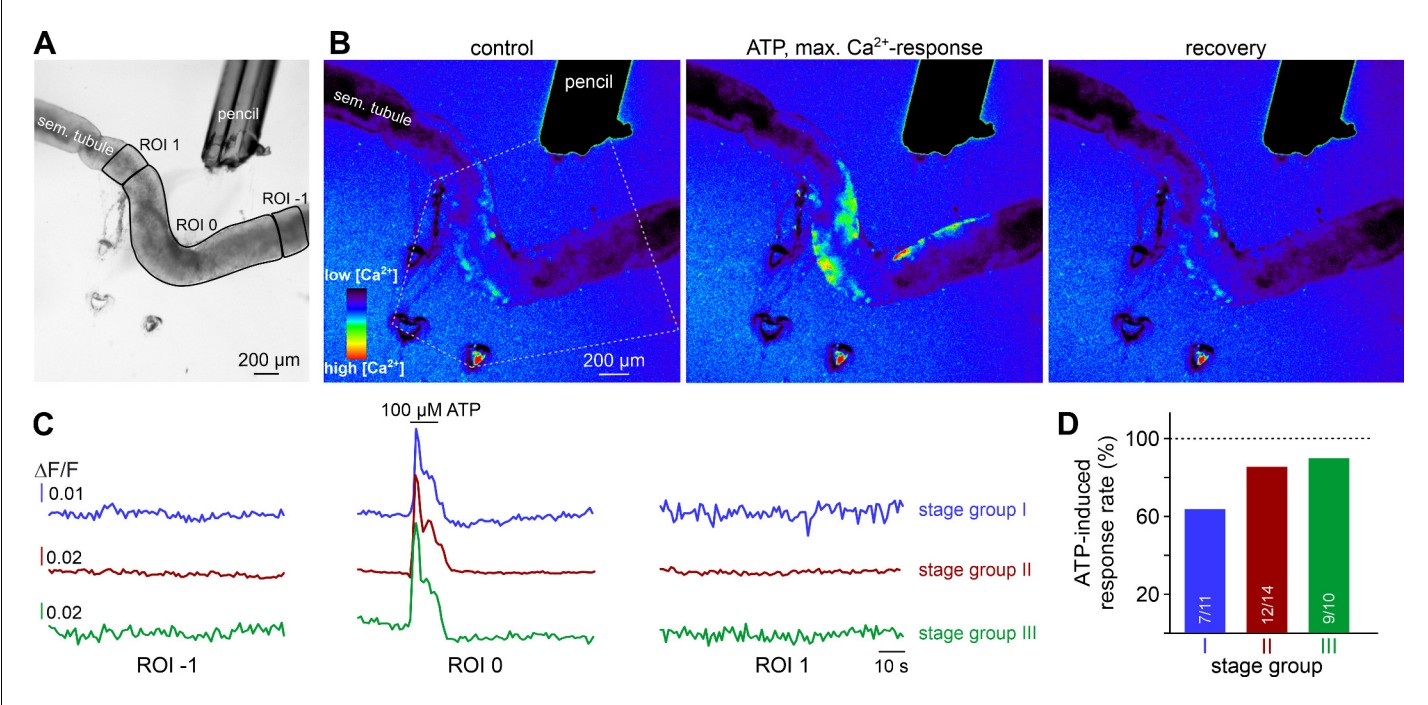

**Figure 6.** ATP causes Ca²⁺ elevations within a restricted paracrine radius. (**A**) Low-magnification brightfield image of an intact seminiferous tubule segment dissected from SMMHC-CreER$^{T2}$ x Ai95D mice and positioned directly in front of the tip of a 250 μm diameter perfusion pencil. ROIs (black lines) are drawn to encompass the area that is directly exposed to fluid flow (ROI 0) as well as adjacent regions (ROIs 1 and −1), respectively. Suction produced by negative pressure (applied through holes in the elastic foil pad beneath the tubule) limits the area of perfusion. (**B**) Pseudocolor GCaMP6f fluorescence intensity images of the tubule shown in (**A**) reveals Ca²⁺ transients in TPCs in response to ATP. Representative images (rainbow 256 color map) correspond to time points before, during, and after focal ATP exposure (100 μM; 10 s). The area directly challenged with ATP is denoted by the white dotted lines. For clarity, autofluorescence of the perfusion pencil was removed. Note that Ca²⁺ elevations are limited to ROI 0. (**C**) Representative original recordings of changes in GCaMP6f intensity (ΔF/F) over time from tubule segments of the three different stage groups (I–III). Traces exemplify Ca²⁺ signals (or the lack thereof) in ROIs 0, −1, and 1, respectively. Independent of the epithelial cycle stage investigated, ATP-induced $[Ca^{2+}]_c$ elevations are restricted to directly exposed tissue segments. (**D**) Quantification of ATP sensitivity among tubule segments of different cycle stage. Bar charts illustrate that purinergic stimulation causes Ca²⁺ signals irrespective of stage and, thus, luminal sperm count. Numbers of experiments as indicated in bars.

flow propagates beyond the directly stimulated tubule section. When we analyzed luminal motion in equidistant tubule sections adjacent to the directly stimulated area ROI 0 (*Figure 5A*), we found a significant, though relatively small bidirectional wave of propagating movement in stage groups I and III, which exhibit a low luminal sperm count (*Figure 5C$_{II}$*). Strikingly, we observed strong unidirectional luminal movement upon ATP stimulation of stage group II tubule sections which show high luminal sperm density associated with spermiation (*Figure 5C$_{II}$*). In this stage group, luminal content is predominantly propelled toward areas of ascending spermatogenic cycle stages. These findings demonstrate directionality of sperm transport upon focal purinergic TPC stimulation in isolated seminiferous tubules.

As expected, ATP-induced tubule contractions also manifest as Ca²⁺ signals in TPCs (*Figure 6A&B*, *Video 5*). However, these Ca²⁺ elevations appear to be limited to those areas directly exposed to ATP (ROI 0). We observed no such signals in adjacent tubule sections independent of the stimulated stage group or an ascending or descending stage direction (*Figure 6C&D*). This finding indicates that, in the isolated seminiferous tubule, ATP acts as a local messenger that, by itself, is not sufficient to trigger a signal that propagates in a regenerative wave-like fashion along a tubule's longitudinal axis. However, local contractions generate sufficient force to move luminal content beyond the directly stimulated area and, in turn, directionality of flow along short-to-medium distances (≤600 μm; *Figure 5C$_{II}$*) is not critically dependent on peristaltic contractility.

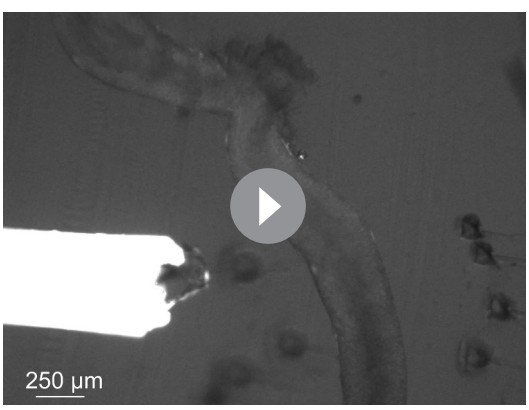

**Video 5.** ATP stimulation triggers transient Ca²⁺ signals in TPCs of intact seminiferous tubules. Fluorescence time-lapse recording of an intact seminiferous tubule (field of view shows cycle stage II) isolated from a mouse selectively expressing GCaMP6f in TPCs (SMMHC-CreER$^{T2}$ x Ai95D male offspring). Fluorescence imaging (ΔF/F) during brief focal ATP perfusion (100 µM, 10 s) – the spatial extent of the stimulation zone had been defined by prior perfusion with a dye solution (food color) – reveals relative changes in TPC Ca²⁺ concentration (rainbow color map; blue, low Ca²⁺; red, high Ca²⁺).
https://elifesciences.org/articles/62885#video5

## ATP induces tubular contractions in vivo

To ultimately attribute a physiological role to ATP-dependent Ca²⁺ signals in TPCs, tubular contractions, and corresponding transport of luminal content, these phenomena must (i) occur spontaneously in living animals, and must (ii) be triggered experimentally by ATP exposure in vivo. Thus, to investigate any in vivo relevance of our findings, we designed a custom-built 3D printed in vivo imaging stage (*Figure 7—figure supplement 1*) that allows both widefield epi-fluorescence and multiphoton microscopy of the mouse testis.

Initially, we monitored spontaneous seminiferous tubule activity in SMMHC-CreER$^{T2}$ x Ai95D mice. Multiphoton time-lapse imaging revealed spontaneous TPC Ca²⁺ signals that typically accompanied strong tubule contractions (*Figure 7A&B*, *Video 6*). Several characteristics emerged from quantitative analysis of these observations. First, during sufficiently long recording periods (≤30 min), contractions occur in essentially all seminiferous tubules (*Figure 7— figure supplement 2A*). Second, contractions of individual tubules within the 2D confocal plane are not synchronized (*Figure 7B*). Third, periods of enhanced activity (≥2 contractions within 90 s) are interrupted by long episodes of quiescence (*Figure 7B*, *Figure 7—figure supplement 2B*). Fourth, the durations of TPC Ca²⁺ signals and corresponding contractions are positively correlated (*Figure 7—figure supplement 2C*), confirming a causal relationship.

Next, we asked whether spontaneous in vivo contractions are coordinated along the longitudinal tubular axis. Low magnification incident light microscopy enabled simultaneous observation of several superficial seminiferous tubule segments (*Figure 7C*). Movement analysis along the length of digitally straightened tubules demonstrates wave-like unidirectional motions that propagate with high velocities (*Figure 7C&D*). These movements coincide with 'macroscopic' Ca²⁺ waves that travel at comparable speed and direction (*Figure 7—figure supplement 2D*). Notably, the observed coordinated contractile movements provide sufficient force to ensure luminal sperm transport (*Video 7*).

Finally, we examined if brief focal ATP stimulation also triggers peritubular Ca²⁺ signals and seminiferous tubule contractions in vivo. Therefore, we filled low resistance patch pipettes with fluorescently labeled ATP solution, penetrated the tunica albuginea, and targeted the interstitial space close to neighboring tubules (*Figure 7E*, *Video 8*). Nanoliter puffs of ATP-containing test solution induced both Ca²⁺ transients in genetically labeled TPCs and strong tubule contractions in the majority of experiments (*Figure 7F&G*). By contrast, puffs of extracellular saline rarely stimulated any such response (*Figure 7—figure supplement 2E*). Taken together, in vivo recordings demonstrate that robust recurrent seminiferous tubule contractions (i) occur spontaneously, (ii) are driven by cytosolic Ca²⁺ elevations in TPCs that propagate in a wave-like fashion, and (iii) can be triggered experimentally by ATP exposure. Consequently, paracrine purinergic signaling in the mouse testis is a mediator of luminal sperm transport within the seminiferous tubule network.

## Discussion

The molecular and cellular mechanisms that control paracrine testicular communication have to a large extent remained controversial, if not elusive (*Schlatt and Ehmcke, 2014*). For TPCs in particular, a contractile function under paracrine control and, consequently, a critical role in male infertility have long been proposed (*Albrecht et al., 2006*; *Romano et al., 2005*), but direct experimental

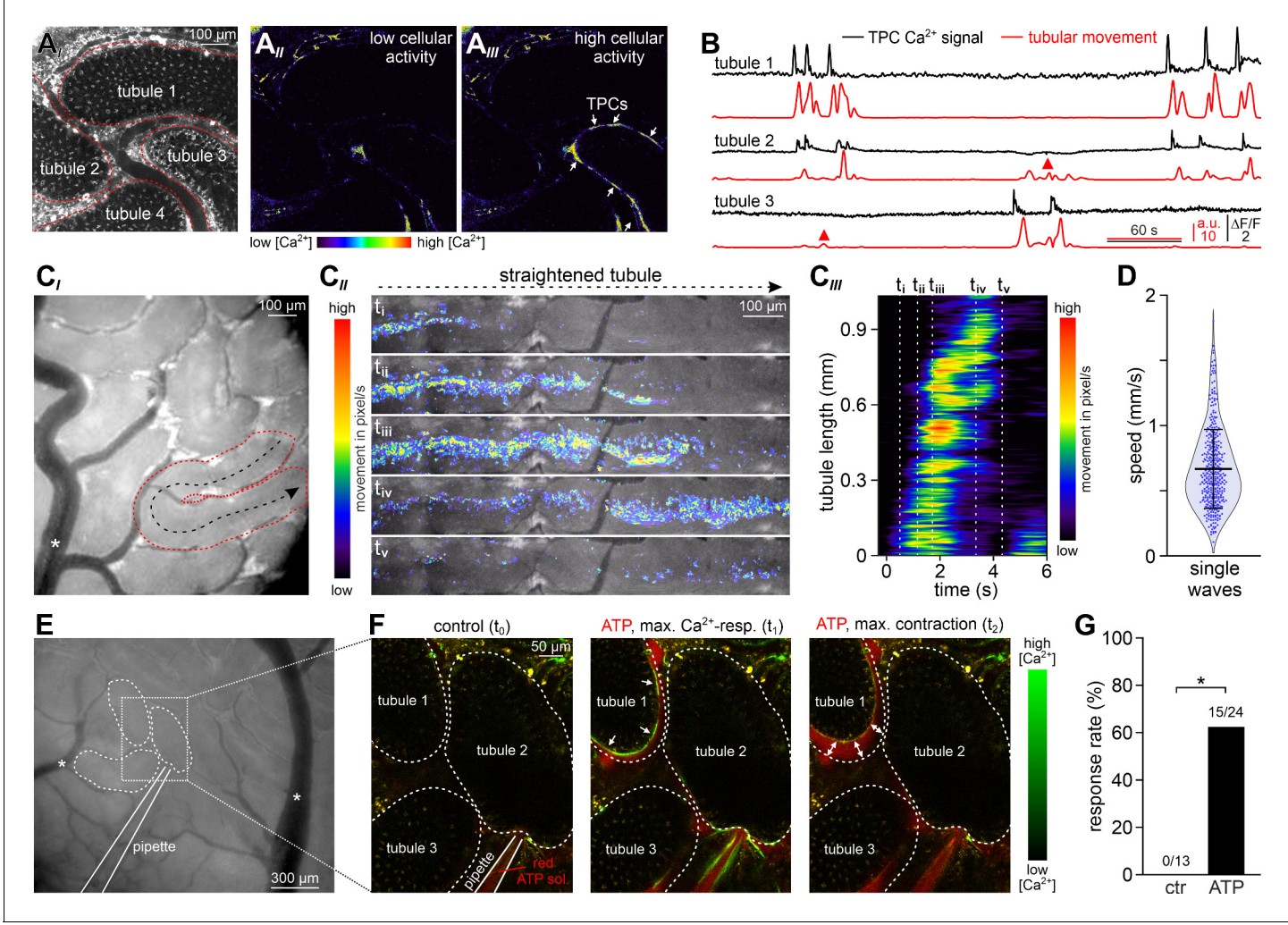

**Figure 7.** ATP induces tubular contractions in vivo. (A) Multiphoton in vivo fluorescence microscopy in SMMHC-CreER[T2] x Ai95D mice enables time-lapse imaging of TPC activity. Maximum gray scale projection outlines segments from four seminiferous tubules (red dotted lines ($A_I$)). Pseudocolor images of GCaMP6f intensity indicate $[Ca^{2+}]_c$ changes in TPCs of tubule three during phases of low ($A_{II}$) vs. high ($A_{III}$) spontaneous activity (rainbow color map; white arrows in ($A_{III}$)). (B) Original traces depict simultaneous TPC $Ca^{2+}$ signals (black; ΔF/F) and tubular contractions (red; calculated as flow change $c_i$ relative to each previous image (methods) over time in tubules 1–3) (A). Red triangles mark passive movements, which occur upon contractions of adjacent tubules. Note the lack of a corresponding $Ca^{2+}$ signal. (C) Analysis of spontaneous tubular motion in vivo. Low magnification incident light image of the mouse testis ($C_I$) shows several superficial seminiferous tubule segments, testicular blood vessels (white asterisk; note that unobstructed blood supply (i.e. visualizing erythrocyte flow) is checked routinely), and a specific segment outlined by red dotted lines. After time-lapse imaging, this segment is digitally straightened ($C_{II}$) and subjected to motion analysis. For different time points (i–v), pixel movement and its propagation are reflected by merged pseudocolor images. Directionality is indicated by the black arrow in ($C_I$). From a kymograph ($C_{III}$), the time–space relationship of tubular motion becomes apparent (time points i–v as indicated by dashed vertical lines). (D) Violin plot depicting the velocity of contractile movement in individual tubule segments (blue dots). (E–G) ATP-induced $Ca^{2+}$ signals and contractions in vivo. (E) Low magnification epi-fluorescence image of several superficial seminiferous tubule segments and blood vessels (white asterisks). The boxed area includes three tubule segments (dotted black lines), which are targeted by a low resistance pipette filled with fluorescently labeled ATP solution. (F) Enlarged view of the area outlined in (E). Merged (red/green) multiphoton fluorescence images taken before and during/after brief stimulation with ATP. The middle and right frames correspond to the point of maximum $Ca^{2+}$ signal (green) and contraction (double arrows) of tubule 1, respectively. (G) Bar chart quantification of contractions induced by nanoliter puffs of saline with or without ATP (1 mM). Asterisk denotes statistical significance (p=0.036; Fisher's Exact test). The online version of this article includes the following figure supplement(s) for figure 7:

**Figure supplement 1.** A custom-built 3D printed microscope stage enables simultaneous in vivo multiphoton imaging of $Ca^{2+}$ signals and contractions in mouse seminiferous tubules.

**Figure supplement 2.** In vivo imaging of tubular activity in SMMHC-CreER[T2] x Ai95D mice.

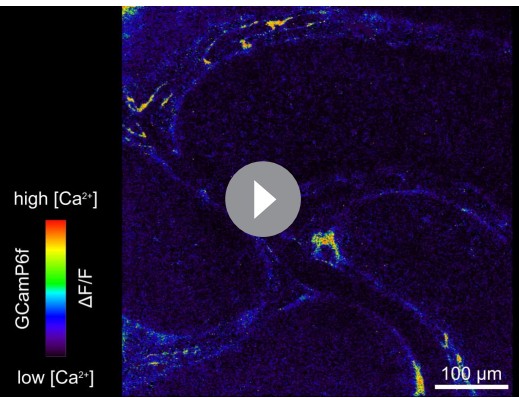

**Video 6.** In vivo multiphoton microscopy demonstrates spontaneous $Ca^{2+}$ signals in mouse TPCs. Spontaneous seminiferous tubule in vivo activity monitored in SMMHC-CreER$^{T2}$ x Ai95D mice. Intravital multiphoton fluorescence time-lapse imaging ($\Delta F/F$, 2 Hz) reveals coordinated changes in TPC $Ca^{2+}$ concentration (rainbow color map; blue, low $Ca^{2+}$; red, high $Ca^{2+}$) among one of three seminiferous tubules in the field of view (591 µm x 591 µm).

https://elifesciences.org/articles/62885#video6

release as well as the mechanism(s) that trigger ATP secretion in vivo currently remain elusive. The apparent absence of efferent nerve endings in the seminiferous tubules and interstitial tissue (*Tripiciano et al., 1996*) suggests that tubule contractility is under endo-/paracrine control. By

evidence has been lacking (*Mayerhofer, 2013*). While several signaling molecules, including vasopressin (*Pickering et al., 1989*), oxytocin (*Worley et al., 1985*), prostaglandins (*Hargrove et al., 1975*), endothelin (*Filippini et al., 1993*), and others (*Albrecht et al., 2006*; *Mayerhofer, 2013*), have been proposed to act on TPCs, a role of ATP in seminiferous tubule contractility has been explicitly ruled out early on (*Hovatta, 1972*). By contrast, our data reveal ATP is a strong stimulus that activates TPCs via both P2X and P2Y receptors, mediating coordinated tubule contractions and luminal sperm transport in situ and in vivo. Both spontaneous and ATP-dependent contractions trigger fast, stage-dependent, and directional transport of luminal content. It is thus tempting to speculate that seminiferous tubule contractility in general, and purinergic TPC signaling in particular, are promising targets for male infertility treatment and/or contraceptive development.

The site(s)/cellular origin of testicular ATP

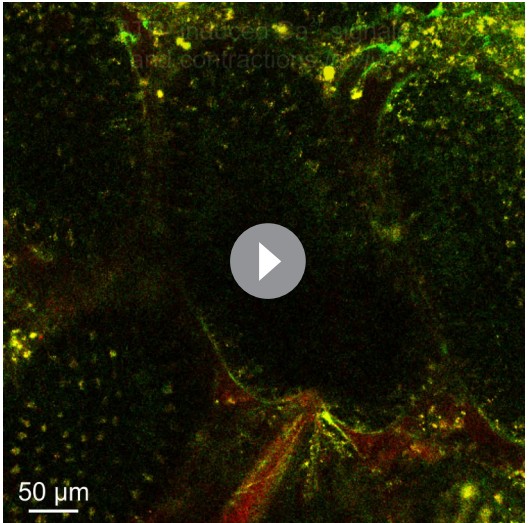

**Video 8.** Focal ATP stimulation triggers peritubular $Ca^{2+}$ signals and seminiferous tubule contractions in vivo. Intravital multiphoton fluorescence time-lapse imaging in SMMHC-CreER$^{T2}$ x Ai95D mice. Overlay of two detection channels ($\Delta F/F$, GCaMP6f, green; Alexa Fluor 555, red). Stimulus solution (containing Alexa Fluor 555 (4 µM) and ATP (1 mM)) is puffed from a glass micropipette, which penetrated the *tunica albuginea* to target the interstitial space. Changes in TPC $Ca^{2+}$ concentration are color-coded (black, low $Ca^{2+}$; green, high $Ca^{2+}$). Note that typically such contractions / $Ca^{2+}$ signals do not occur when ATP is omitted from the 'puff' solution (data not shown).

https://elifesciences.org/articles/62885#video8

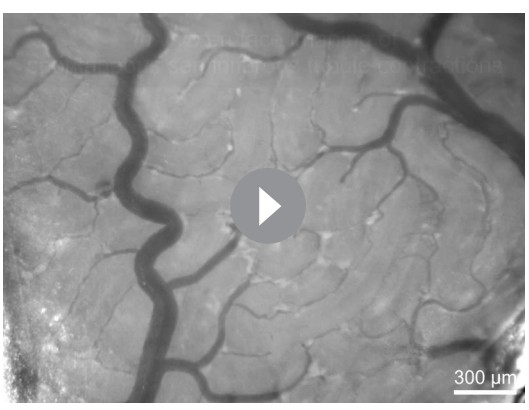

**Video 7.** Coordinated contractile movements ensure luminal sperm transport in vivo. Intravital en-face brightfield imaging illustrates spontaneous contractions and luminal movement in seminiferous tubules of adult mice. Low (1.5 x 1.4 mm field of view) and high-magnification time-lapse recordings reveal that contractions and luminal content propulsion are routinely observed in vivo. Note the unobstructed blood flow within testicular vessels.

https://elifesciences.org/articles/62885#video7

contrast, autonomic innervation of the testicular capsule mediates smooth muscle cell contraction of the *tunica albuginea*, using ATP as a (co)transmitter (*Banks et al., 2006*). Regulated ATP release has been reported for both Sertoli and germ cells (*Gelain et al., 2005*; *Gelain et al., 2003*). Moreover, TPCs express P2Y6 receptors (*Figure 1C*), which were reported to mediate ATP release upon activation (*Carneiro et al., 2014*). Thus, TPCs could themselves participate in regenerative nucleotide release.

During spermatogenesis, apoptosis is a vital process (*Print and Loveland, 2000*). In fact, up to 75% of germ cells undergo apoptosis under physiological conditions (*Huckins, 1978*). This substantial germ cell loss is called 'density-dependent regulation' (*Hess and De Franca, 2008*). Since ATP release from apoptotic cells is well documented (*Elliott et al., 2009*) it is likely that cell density-dependent waves of apoptosis could regularly generate local ATP surges. We have previously shown that one result of seminiferous extracellular ATP elevation is signal amplification by increased ATP release (*Fleck et al., 2016*), although the mechanistic basis of this positive feedback pathway is yet unknown. Given (i) the robust cytosolic $Ca^{2+}$ transients observed in response to ATP exposure in various testicular cell types (*Fleck et al., 2016*; *Gelain et al., 2005*; *Liévano et al., 1996*; *Veitinger et al., 2011*; *Walenta et al., 2018*) and (ii) the usually millimolar ATP content in secretory vesicles (*Bodin and Burnstock, 2001*; *Zhang et al., 2007*), the most parsimonious explanation for ATP-induced ATP release would be an inevitable ATP 'co-secretion' upon any $Ca^{2+}$-dependent exocytosis event. In addition, ATP release has been observed in several cell types as a result of mechanical deformation, shear stress, stretch, or osmotic swelling (*Button et al., 2013*) adding another putative mechanism of regenerative signaling in purinergic contraction control.

Notably, extracellular ATP is rapidly degraded by ecto-nucleotidases (*Zimmermann et al., 2012*), rendering its interstitial half-life relatively short and, thus, narrowing its paracrine radius to a few hundred micrometers (*Fitz, 2007*). Combined with its fast diffusion – approximately 1 μm in less than 10 ms (*Khakh, 2001*) – extracellular ATP bears all characteristics of a fast paracrine agent in testicular communication (*Praetorius and Leipziger, 2009*).

Excitation–contraction coupling in TPCs is poorly described. Our results strongly suggest that a combination of P2X (isoforms 2 and/or 4) receptor-dependent $Ca^{2+}$ influx and P2Y (likely isoform 2) receptor-activated phospholipase C$\beta$-dependent $Ca^{2+}$ release from the sarcoplasmic reticulum – the latter being critical and resembling the recently reported mechanism of vascular smooth muscle cell contraction in small pulmonary veins (*Henriquez et al., 2018*) – provides the $[Ca^{2+}]_c$ elevation required for force generation (*Berridge, 2008*). P2X and P2Y receptors act on different time scales and display different ligand sensitivity, with $EC_{50}$ values in the nanomolar (P2Y) vs. micromolar (P2X) range (*North, 2002*). It is possible that the co-activation of an ionotropic (P2X) and a metabotropic (P2Y) signaling pathway serves functions analogous to the concomitant exposure to both ATP and noradrenaline in mesenteric artery smooth muscle. Here, activation of P2X1 receptors generates a small initial contraction that is followed by larger noradrenaline-induced contraction (*Lamont et al., 2006*; *Lamont et al., 2003*). Regarding TPC $[Ca^{2+}]_c$ elevation, our data suggest that P2X receptor activation also targets $Ca^{2+}$ release from internal stores, as their depletion inhibits excitation–contraction coupling entirely. Therefore, it is likely that P2X receptors act as signal boosters that mediate $Ca^{2+}$-induced $Ca^{2+}$ release, possibly via activation of ryanodine receptors (*Berridge, 2008*). This way, the combined action of P2X and P2Y receptors might equip TPCs with a broader 'two-step' stimulus integration range.

Whole-mount imaging of isolated seminiferous tubules reveals propagation of luminal content that extends beyond the confines of the stimulated/contracted area and displays stage-dependent directionality. While peristaltic contractions are driven by propagating wave-like $Ca^{2+}$ signals in vivo, focal ATP stimulation appears insufficient to trigger a regenerative $Ca^{2+}$ wave in isolated tubules. We, thus, conclude that the observed directionality results from other, likely structural characteristics, for example anatomical features of stage group II and III tubules that favor a specific flow direction (increased tubule diameter and reduced luminal resistance along the stage II-to-III transition zone). We cannot, however, rule out that the use of large field-of-view/low numerical aperture objectives for 'macroscopic' imaging simply prevents the detection of low-amplitude $Ca^{2+}$ signal spread.

Translation of our findings from the mouse model to humans awaits further in-depth investigation. We have recently reported that ATP activates $Ca^{2+}$ signals in human TPCs in vitro (*Walenta et al., 2018*). Moreover, our present findings reveal ATP-induced contractions in cultured human TPCs. There are, however, notable differences between the human and the mouse tubular walls. While a

single layer of TPCs surrounds the mouse seminiferous tubules, the human tubular wall architecture is more complex, containing several TPC layers, substantial amounts of extracellular matrix proteins, and immune cells (*Mayerhofer, 2013*). Impaired spermatogenesis in sub-/infertile men typically coincides with tubular wall remodeling and a partial loss of TPC contractility proteins has been reported in infertile men (*Welter et al., 2013*). Accordingly, interference with TPC contractility had been proposed as a promising strategy for human male contraception (*Romano et al., 2005*). However, a causal relationship between contractility (or the lack thereof) and male (in)fertility has never been established. In fact, seminiferous tubule contractions had, so far, never been observed in vivo and most in vitro reports were based on indirect and non-quantitative evidence, for example from *post-hoc* fluorescence or scanning electron microscopy (*Barone et al., 2002*; *Fernández et al., 2008*; *Losinno et al., 2016*; *Losinno et al., 2012*; *Tripiciano et al., 1999*; *Tripiciano et al., 1997*; *Tripiciano et al., 1996*), morphometry of single cells in culture (*Rossi et al., 2002*; *Santiemma et al., 2001*; *Santiemma et al., 1996*; *Tripiciano et al., 1996*), or intraluminal pressure analysis (*Miyake et al., 1986*; *Yamamoto et al., 1989*). The fact that expression of TPC contractility proteins initiates with puberty under androgen control and that selective androgen receptor knockout in TPCs renders mice infertile (*Welsh et al., 2009*) underscores a potential role of TPC contractions in male fertility. Accordingly, pharmacological targeting of purinergic signaling pathways to (re)gain control of TPC contractility represents an attractive approach for male infertility treatment or contraceptive development.

Among several remaining questions, future experimental efforts will have to address (i) whether TPCs are coupled by gap junctions to display coordinated activity; (ii) whether and, if so, how the final ATP metabolite adenosine affects seminiferous tubule physiology; (iii) whether Rho/Rho kinase signaling pathways modulate TPC contractility as frequently observed in other smooth muscle cells (*Somlyo and Somlyo, 2003*); (iv) what, if any, role is played by P2X receptor-dependent changes in membrane potential; (v) which function is served by the sustained $Ca^{2+}$-gated $Cl^-$ current (*Figure 1—figure supplement 2*); (vi) why periods of enhanced contractile activity are interrupted by longer quiescent episodes (*Figure 7—figure supplement 2B*); (vii) which additional or complementary roles in TPC physiology are played by previously proposed activators, including vasopressin, oxytocin, prostaglandins, and endothelin (*Albrecht et al., 2006*; *Mayerhofer, 2013*); (viii) whether an additional cytosolic and/or membrane $Ca^{2+}$ oscillator (*Berridge, 2008*) provides an endogenous pacemaker mechanism that acts independent of purinergic stimulation; and (ix) whether, similar to vascular smooth muscle cells, some specific tone is maintained between contractions by spatial averaging of asynchronous oscillations (*Berridge, 2008*), a mechanism that could explain the occurrence of spontaneous low-amplitude 'vibratory' movements and local indentations that we (*Figure 4A*) and others (*Ellis et al., 1981*; *Worley et al., 1985*) have observed.

# Materials and methods

## Key resources table

| Reagent type (species) or resource | Designation | Source or reference | Identifiers | Additional information |
|---|---|---|---|---|
| Strain, strain background (*M. musculus*) | C57BL/6J | Charles River Laboratories | Jax # 000664, RRID:IMSR_JAX:000664 | |
| Strain, strain background (*M. musculus*) | SMMHC-CreERT2 | Jackson Laboratories | Jax # 019079, RRID:IMSR_JAX:019079 | |
| Strain, strain background (*M. musculus*) | 129S.FVB-Tg(Amh-cre) 8815Reb/J | Jackson Laboratories | Jax # 007915, RRID:IMSR_JAX:007915 | |
| Strain, strain background (*M. musculus*) | Ai95D | Jackson Laboratories | Jax # 028865, RRID:IMSR_JAX:028865 | Cre-dependent GCaMP6f expression |
| Strain, strain background (*M. musculus*) | Ai14D | Jackson Laboratories | Jax # 007914, RRID:IMSR_JAX:007914 | Cre-dependent tdTomato expression |

*Continued on next page*

*Continued*

| Reagent type (species) or resource | Designation | Source or reference | Identifiers | Additional information |
|---|---|---|---|---|
| primary cells (*M. musculus*) | testicular peritubular cells (TPC) | this paper | | early passage number, Spehr laboratory (see: **TPC culture**) |
| primary cells (*Homo sapiens*) | testicular peritubular cells (TPC) | this paper | | early passage number *Albrecht et al., 2006*, Mayerhofer laboratory |
| Biological sample (*M. musculus*) | seminiferous tubules | this paper | | freshly isolated from *Mus musculus*, Spehr laboratory (see: **Slice preparation**) |
| Antibody | anti-actin, α-smooth muscle - FITC antibody (α-SMA-FITC); mouse, monoclonal | Millipore Sigma | cat # F3777, RRID:AB_476977 | (1:500) |
| Sequence-based reagent | P2 receptors | this paper | PCR primers | table in methods section (see: **Gene expression analysis**), Spehr laboratory |
| Commercial assay, kit | RevertAid H Minus kit | Thermo Fisher | cat # K1632 | |
| Chemical compound, drug | soybean trypsin inhibitor (SBTI) | Sigma Aldrich | cat # T6522 | (100 µg/ml) |
| Chemical compound, drug | fura-2/AM | Thermo Fisher Scientific | cat # F-1201 | (cell culture: 5 µM, tissue slices: 30 µM) |
| Chemical compound, drug | ivermectin | Sigma Aldrich | cat # I8898 | (3 µM) |
| Chemical compound, drug | suramin | Sigma Aldrich | cat # S2671 | (100 µM) |
| Chemical compound, drug | cyclopiazonic-acid (CPA) | Tocris Bioscience | cat # 1235 | (90 µM) |
| Software, algorithm | Imaris 8 | Bitplane | RRID:SCR_007370 | microscopy image analysis software |
| Software, algorithm | custom-written MATLAB code | this paper | https://github.com/rwth-lfb/Fleck_Kenzler_et_al; *Fleck, 2021*; copy archived at swh:1:rev:88c8792860ddf09fd7da969fef6bf86c40441135 | contraction and Ca$^{2+}$ signal analysis, Merhof laboratory (see: **Data analysis**) |
| Other | DAPI stain | Thermo Fisher Scientific | cat # D1306, RRID:AB_2629482 | (5 µg/ml) |
| Other | DRAQ5 stain | Thermo Fisher Scientific | cat # 65-0880-96, RRID:AB_2869620 | (IH: 1:500 CLARITY: 1:1000) |

## Animals

All animal procedures were approved by local authorities and in compliance with both European Union legislation (Directive 2010/63/EU) and recommendations by the Federation of European Laboratory Animal Science Associations (FELASA). When possible, mice were housed in littermate groups of both sexes (room temperature (RT); 12:12 hr light-dark cycle; food and water available ad libitum). If not stated otherwise, experiments used adult (>12 weeks) males. Mice were killed by $CO_2$ asphyxiation and decapitation using sharp surgical scissors. We used C57BL/6J mice (Charles River Laboratories, Sulzfeld, Germany) as well as offspring from crossing either SMMHC-CreER$^{T2}$ (JAX #019079) (*Wirth et al., 2008*) or 129S.FVB-Tg(Amh-cre)8815Reb/J (JAX #007915) (*Holdcraft and Braun, 2004*) mice with either Ai95D (JAX #028865) (*Madisen et al., 2015*) or Ai14D (JAX #007914) (*Madisen et al., 2010*) mice, respectively.

## Chemicals and solutions

The following solutions were used:

($S_1$) 4-(2-Hydroxyethyl)piperazine-1-ethanesulfonic acid (HEPES) buffered extracellular solution containing (in mM) 145 NaCl, 5 KCl, 1 $CaCl_2$, 0.5 $MgCl_2$, 10 HEPES; pH = 7.3 (adjusted with NaOH); osmolarity = 300 mOsm (adjusted with glucose).

($S_2$) Oxygenated (95% $O_2$, 5% $CO_2$) extracellular solution containing (in mM) 120 NaCl, 25 $NaHCO_3$, 5 KCl, 1 $CaCl_2$, 0.5 $MgCl_2$, 5 N,N-bis(2-hydroxyethyl)−2-aminoethanesulfonic acid (BES); pH = 7.3; 300 mOsm (glucose).

($S_3$) Extracellular low $Ca^{2+}$ solution containing (in mM) 145 NaCl, 5 KCl, 0.5 $MgCl_2$, 10 HEPES; pH = 7.3 (NaOH); osmolarity = 300 mOsm (glucose); $[Ca^{2+}]_{free}$ = ~110 nM (1 mM EGTA, 0.5 mM $CaCl_2$) or ~12 nM (1 mM EGTA, 0.1 mM $CaCl_2$).

($S_4$) Oxygenated (95% $O_2$, 5% $CO_2$) extracellular solution containing (in mM) 120 NaCl, 25 $NaHCO_3$, 5 KCl, 0.5 $MgCl_2$, 5 BES; pH = 7.3; 300 mOsm (glucose); $[Ca^{2+}]_{free}$ = ~110 nM (1 mM EGTA, 0.5 mM $CaCl_2$) or ~12 nM (1 mM EGTA, 0.1 mM $CaCl_2$).

($S_5$) Gluconate-based extracellular solution containing (in mM) 122.4 Na gluconate, 22.6 NaCl, 5 KCl, 1 $CaCl_2$, 0.5 $MgCl_2$, 10 HEPES; pH = 7.3 (adjusted with NaOH); osmolarity = 300 mOsm (glucose).

($S_6$) Standard pipette solution containing (in mM) 143 KCl, 2 KOH, 1 EGTA, 0.3 $CaCl_2$, 10 HEPES ($[Ca^{2+}]_{free}$ = ~110 nM); pH = 7.1 (adjusted with KOH); osmolarity = 290 mOsm (glucose).

($S_7$) Gluconate-based pipette solution containing (in mM) 110 Cs gluconate, 30 CsCl, 2 CsOH, 1 EGTA, 0.3 $CaCl_2$, 10 HEPES ($[Ca^{2+}]_{free}$ = ~110 nM); pH = 7.1 (adjusted with CsOH); osmolarity = 290 mOsm (glucose).

In some experiments Na-GTP (0.5 mM) was added to the pipette solution. Free $Ca^{2+}$ concentrations were calculated using WEBMAXCLITE v1.15 (RRID:SCR_000459). If not stated otherwise, chemicals were purchased from Sigma (Schnelldorf, Germany). Cyclopiazonic-acid (CPA) and 2'(3')-O-(4-Benzoylbenzoyl)adenosine-5'-triphosphate (BzATP) triethylammonium salt was purchased from Tocris Bioscience (Bristol, UK). Fura-2/AM was purchased from Thermo Fisher Scientific (Waltham, MA). Final solvent concentrations were ≤0.1%. When high ATP concentrations (≥1 mM) were used, pH was readjusted.

## Stimulation

For focal stimulation, solutions and agents were applied from air pressure-driven reservoirs via an 8-in-1 multi-barrel 'perfusion pencil' (AutoMate Scientific; Berkeley, CA). Changes in focal superfusion (*Veitinger et al., 2011*) were software-controlled and, if required, synchronized with data acquisition by TTL input to 12V DC solenoid valves using a TIB 14S digital output trigger interface (HEKA Elektronik, Lambrecht/Pfalz, Germany). For focal stimulation during in vivo recordings, ATP was puffed from pulled glass pipettes using a microinjection dispense system (Picospritzer III; Parker Hannifin, Hollis, NH).

Low $[Ca^{2+}]_e$ solutions ($S_3$ and $S_4$) were applied via both the bath and perfusion pencil. To ensure depletion of $Ca^{2+}$ stores by CPA we monitored intracellular $Ca^{2+}$ levels during drug treatment (0.05 Hz frame rate). Transient CPA-dependent $Ca^{2+}$ elevations lasted 10–40 min. After baseline $Ca^{2+}$ levels were restored, cells/slices were again challenged with ATP. Control recordings, omitting CPA, were performed under the same conditions.

## Slice preparation

Acute seminiferous tubule slices were prepared as previously described (*Fleck et al., 2016*) with minor modifications. Briefly, seminiferous tubules from young adults were isolated after *tunica albuginea* removal, embedded in 4% low-gelling temperature agarose (VWR, Erlangen, Germany), and 250 µm slices were cut with a VT1000S vibratome (RRID:SCR_016495; Leica Biosystems, Nussloch, Germany). Acute slices were stored in a submerged, oxygenated storage container ($S_2$; RT). When using testicular tissue from Ai95D mice, slices were protected from light during storage to avoid GCaMP6f bleaching.

## TPC culture

After mouse testis isolation and removal of the *tunica albuginea*, the seminiferous tubules were placed in Dulbecco's Modified Eagle Medium/Nutrient Mixture F-12 (DMEM/F-12; Invitrogen) containing 1 mg $ml^{-1}$ collagenase A and 6 µg $ml^{-1}$ DNase (10 min; 34°C; shaking water bath (60 cycles

min$^{-1}$)). Three times, the samples were washed (DMEM/F-12; 5 ml), allowed to settle for 5 min, and the supernatant was discarded. Next, tubules were incubated DMEM/F-12 containing 1 mg ml$^{-1}$ trypsin and 20 µg ml$^{-1}$ DNase (20 min; 34˚C; shaking water bath (60 cycles min$^{-1}$)). Digestion was stopped by addition of 100 µg ml$^{-1}$ soybean trypsin inhibitor (SBTI) and 20 µg ml$^{-1}$ DNase in phosphate-buffered saline (D-PBS). Then, samples were allowed to settle for 5 min and the supernatant was collected. After two more cycles of washing (DMEM/F-12), settling (5 min), and supernatant collection, the collected cell suspension was centrifuged (10 min; 400 g) and the supernatant discarded. The pellet was resuspended in DMEM containing FBS (10%) and penicillin G/streptomycin (1%), filtered (cell strainer (100 µm)), and cells were plated in 75 cm$^2$ cell culture flask (T75; Invitrogen) and placed in a humidified incubator (37˚C; 5% CO$_2$). Approximately $^1/_3$ of medium volume was replaced every 3 days. Cells usually reached 100% confluence after 7 days in vitro (DIV). Then, cells were washed twice (DPBS$^{-/-}$; 5 min; 37˚C) and incubated in 0.05% trypsin/EDTA (5 min; 37˚C). Detachment of cells was checked visually and, if necessary, facilitated mechanically. The cell suspension was centrifuged (3 min; 800 g) and the supernatant discarded. The pellet was resuspended in DMEM at cell densities of ~10$^5$ cells ml$^{-1}$ and plated again either in culture flasks or on glass coverslips in 35 mm dishes for experimental use. Again, $^1/_3$ of medium volume was replaced every 3 days. Experiments were performed for ≤5 days after passage.

Human TPCs were isolated from small testicular tissue fragments derived from consenting donors with obstructive azoospermia and normal spermatogenesis as described (*Albrecht et al., 2006*; *Walenta et al., 2018*). The study was approved by the local ethical committee (Ethikkommission, School of Medicine, TU Munich, project 169/18S).

## Gene expression analysis

Total RNA was isolated and purified from cultured mouse TPCs (passage 1) with Trizol followed by complementary DNA synthesis with RevertAid H Minus kit (#K1632 Thermo Fisher) according to the manufacturer's instructions. Controls in which the reverse transcriptase was omitted were routinely performed. PCR amplification was performed during 30 thermal cycles (95˚C, 20 s; 58˚C, 20 s; 72˚C, 20 s). The following specific primer pairs were used for PCR amplification:

| Target | Forward primer 5´–3´ | Reverse primer 5´–3´ |
| --- | --- | --- |
| P2X1 | CCGAAGCCTTGCTGAGAA | GGTTTGCAGTGCCGTACAT |
| P2X2 | GACCTCCATCGGGGTGGGCT | TGGGGTCCGTGGATGTGGAGT |
| P2X3 | CTGCCTAACCTCACCGACAAG | AATACCCAGAACGCCACCC |
| P2X4 | CCCTTTGCCTGCCCAGATAT | CCGTACGCCTTGGTGAGTGT |
| P2X5 | GCTGCCTCCCACTGCAACCC | AAGCCCCAGCACCCATGAGC |
| P2X6 | CCCAGAGCATCCTTCTGTTCC | GGCACCAGCTCCAGATCTCA |
| P2X7 | CCCAGATGGACTTCTCCGAC | GGACTTAGGGGCCACCTCTT |
| P2Y1 | CGACAGGGTTTATGCCACTT | TCGTGTCTCCATTCTGCTTG |
| P2Y2 | CGTGCTCTACTTCGTCACCA | GACCTCCTGTGGTCCCATAA |
| P2Y4 | ACTGGCTTCTGCAAGTTCGT | AGGCAGCCAGCTACTACCAA |
| P2Y6 | CATTAGCTTCCAGCGCTACC | GCTCAGGTCGTAGCACACAG |
| P2Y12 | CATTGCTGTACACCGTCCTG | AACTTGGCACACCAAGGTTC |
| GAPDH | CAAGGTCATCCATGACAACTTTG | GTCCACCACCCTGTTGCTGTAG |

## Immunochemistry and tissue clearing

For immunochemistry of testicular cryosections, testes were fixed with 4% (w/v) paraformaldehyde (PFA) in PBS$^{-/-}$ (10 mM, pH 7.4; ≥12 hr; 4˚C) and subsequently cryoprotected in PBS$^{-/-}$ containing 30% sucrose (≥24 hr; 4˚C). Samples were then embedded in Tissue Freezing Medium (Leica Biosystems), sectioned at 20 µm on a Leica CM1950 cryostat (RRID:SCR_018061; Leica Biosystems), and mounted on Superfrost Plus slides (Menzel, Braunschweig, Germany). For immunostaining of cultured mouse TPCs, cells were washed (3x; PBS$^{-/-}$), fixed with ice-cold 4% PFA in PBS$^{-/-}$ (20 min; RT),

and washed again (3x; PBS$^{-/-}$). For blocking, sections/cells were incubated in PBS$^{-/-}$ containing Tween-20 (0.1%)/BSA (3%) solution (1 hr; RT). After washing (PBS$^{-/-}$; 2 × 5 min), sections/cells were incubated FITC-conjugated monoclonal anti-actin, α-smooth muscle (α-SMA-FITC, cat # F3777, MilliporeSigma) antibody (1:500 in 3% BSA; 1 hr; RT). Excess antibodies were removed by washing (2 × 5 min PBS$^{-/-}$). For nuclear counterstaining, sections/cells were then incubated in PBS$^{-/-}$ containing either DAPI (5 µg ml$^{-1}$; 10 min; RT; Thermo Fisher Scientific) or DRAQ5 (1:500; 5 min; RT; Thermo Fisher Scientific).

Fluorescent images were taken using either an inverted microscope (Leica DMI4000B, Leica Microsystems) or an upright fixed stage scanning confocal microscope (TCS SP5 DM6000 CFS; Leica Microsystems) equipped with a 20 × 1.0 NA water immersion objective (HCX APO L; Leica Microsystems). To control for non-specific staining, experiments in which the primary antibody was omitted were performed in parallel with each procedure. Digital images were uniformly adjusted for brightness and contrast using Adobe Photoshop CS6 (Adobe Systems, San Jose, CA, USA).

For testicular tissue clearing we adopted the CLARITY method (*Chung et al., 2013*) with minor modifications (*Gretenkord et al., 2019*). Briefly, testes from adult mice were fixed overnight at 4°C in hydrogel fixation solution containing 4% acrylamide, 0.05% bis-acrylamide, 0.25% VA-044 Initiator, 4% PFA in PBS$^{-/-}$ to maintain structural integrity. After hydrogel polymerization, lipids were removed by incubation in 4% sodium dodecyl phosphate (SDS) solution with 200 mM boric acid (pH 8.5) over periods of two months. Solutions were changed bi-weekly. During the final incubation period, the nuclear marker DRAQ5 (1:1000) was added. After washing (2 d) with PBST (0.1% TritonX), samples were incubated for 24 hr in RIMS80 containing 80 g Nycodenz, 20 mM PS, 0.1% Tween 20, and 0.01% sodium acid. Cleared samples were imaged using a Leica TCS SP8 DLS confocal microscope, equipped with a digital light-sheet module, 552 nm and 633 nm diode lasers, a HC PL FLUOTAR 5x/0.15 IMM DLS objective (observation), a L 1.6x/0.05 DLS objective (illumination), a DLS TwinFlect 7.8 mm Gly mirror cap, and a DFC9000 sCMOS camera. Rendering and three-dimensional reconstruction of fluorescence images was performed using Imaris 8 microscopy image analysis software (Bitplane, Zurich, Switzerland).

## Electrophysiology

Whole-cell patch-clamp recordings were performed as described (*Fleck et al., 2016*; *Veitinger et al., 2011*). Briefly, mouse TPCs were transferred to the stage of an inverse microscope (DMI 4000B, Leica Microsystems), equipped with phase contrast objectives and a cooled CCD camera (DFC365FX, Leica Microsystems). Cells were continuously superfused with solution $S_1$ (~3 ml min$^{-1}$; gravity flow;~23°C). Patch pipettes (~5 MΩ) were pulled from borosilicate glass capillaries with filament (1.50 mm OD/0.86 mm ID; Science Products) on a PC-10 vertical two-step micropipette puller (Narishige Instruments, Tokyo, Japan), fire-polished (MF-830 Microforge; Narishige Instruments) and filled with $S_6$. An agar bridge (150 mM KCl) connected reference electrode and bath solution. An EPC-10 amplifier (RRID:SCR_018399) controlled by Patchmaster 2.9 software (RRID:SCR_000034; HEKA Elektronik) was used for data acquisition. We monitored and compensated pipette and membrane capacitance ($C_{mem}$) as well as series resistance ($R_{series}$). $C_{mem}$ values served as a proxy for the cell surface area and, thus, for normalization of current amplitudes (i.e. current density). Cells displaying unstable $R_{series}$ values were not considered for further analysis. Liquid junction potentials were calculated using JPCalcW software (*Barry, 1994*) and corrected online. Signals were low-pass filtered [analog 3- and 4-pole Bessel filters (–3 dB); adjusted to $1/3$ - $1/5$ of the sampling rate (10 kHz)]. If not stated otherwise, holding potential ($V_{hold}$) was –60 mV.

## Fluorescence Ca$^{2+}$ imaging

Cultured mouse TPCs were imaged as described (*Veitinger et al., 2011*). Briefly, cells were loaded with fura-2/AM in the dark (5 µM; 30 min; RT; $S_1$) and imaged with an upright microscope (Leica DMI6000FS, Leica Microsystems) equipped for ratiometric live-cell imaging with a 150 W xenon arc lamp, a motorized fast-change filter wheel illumination system for multi-wavelength excitation, a CCD camera (DFC365 FX, Leica), and Leica LAS X imaging software. Ten to thirty cells in randomly selected fields of view were viewed at 20x magnification and illuminated sequentially at 340 nm and 380 nm (cycle time 2 s). The average pixel intensity at 510 nm emission within user-selected ROIs was digitized and calculated as the $f_{340}/f_{380}$ intensity ratio.

For parallel recordings of intracellular $Ca^{2+}$ signals and tubular contractions, acute seminiferous tubule slices were bulk-loaded with fura-2/AM in the dark (30 µM; 30 min; RT). After washing (3x; $S_1$), slices were transferred to a recording chamber and imaged with an upright microscope (Leica DMI6000FS, see above). We installed a custom-built reflective shield beneath the recording chamber for parallel monitoring of fluorescence and reflected light. At 1 Hz imaging cycles, we thus recorded two 510 nm fluorescence images (340/380 nm excitation) and a 'pseudo-brightfield' reflected light image that allowed quasi simultaneous analysis of intracellular $Ca^{2+}$ and tubular movement.

To ensure effective store depletion by CPA treatment, we recorded intracellular $Ca^{2+}$ levels during CPA incubation at low frequency to monitor $Ca^{2+}$ release from the ER, but also prevent phototoxicity. Experiments were only conducted if (i) we detected a substantial gradual rise in intracellular $Ca^{2+}$ upon CPA treatment, and if (ii) functional $Ca^{2+}$ extrusion mechanisms ensured that $Ca^{2+}$-dependent fluorescence signals returned to base level. The time-course of this $Ca^{2+}$ release – $Ca^{2+}$ extrusion process varied between samples and ranged between 5.3 and 44.0 min (18.8 ± 9.3 min; mean ± SD).

## Whole-mount seminiferous tubule imaging

Isolated tubules (>1 cm length) were placed onto a membrane within a custom-built 3D printed two-compartment recording chamber that was constantly superfused with $S_1$. Small membrane holes under the tubules and around a defined stimulation area allowed for (i) gentle fixation of the tubules and (ii) focal ATP perfusion of selected tubular regions by vacuum-generated negative pressure (80–180 mmHg) in the submembraneous chamber compartment and continuous suction of $S_1$ from the top compartment. After visual determination of tubular stages (I – III) (*Parvinen, 1982*), the perfusion pencil was positioned to selectively stimulate an area of known and homogeneous stage. Focal stimulation in the desired area was routinely confirmed by transient dye perfusion (Fast Green) prior to ATP exposure. ATP stimulations (100 µM; 10 s) and corresponding negative controls were compared to determine ATP-dependent $Ca^{2+}$ signals (offspring from crossing SMMHC-CreER$^{T2}$ and Ai95D mice) or tubular contractions and sperm transport. For low-magnification brightfield or fluorescence imaging, we used a MacroFluo Z16 APO A system (Leica Microsystems) equipped with either a DFC450C camera and a PLANAPO 1.0x/WD 97 mm objective (brightfield) or with a monochrome DFC365FX camera and a 5.0x/0.50 LWD PLANAPO objective (fluorescence). Images were acquired at 1 Hz.

## In vivo imaging

We administered tamoxifen (75 mg tamoxifen kg$^{-1}$ body weight) to double-positive adult male offspring (SMMHC-CreER$^{T2}$ x Ai95D) via daily intraperitoneal injections for five consecutive days. Mice were closely monitored for any adverse reactions to the treatment. Experiments were performed 2–5 weeks after the first injection. For surgery, mice were anesthetized with ketamine-xylazine-buprenorphine (100, 10, 0.05–0.1 mg kg$^{-1}$, respectively; Reckitt Benckiser Healthcare, UK). First, we made an incision next to the *linea alba* in the hypogastric region, followed by a 5 mm incision into the peritoneum. One testis was gently lifted from the abdominal cavity. Its *gubernaculum* was cut and the testis – with the spermatic cord, its blood vessels and *vas deferens* still intact – was transferred to a temperature-controlled imaging chamber filled with extracellular solution ($S_1$; 35°C), mounted on a custom-designed 3D printed in vivo stage (*Figure 7—figure supplement 1*). Throughout each experiment, vital signs (heartbeat, blood oxygen level, breathing rhythm) were constantly monitored and recorded (breathing). Moreover, we routinely checked unobstructed blood flow within testicular vessels during experiments. To avoid movement artifacts, the tunica was glued to two holding strings using Histoacryl tissue adhesive. After surgery, anesthesia was maintained by constant isoflurane inhalation (1–1.5% in air). Time-lapse intravital imaging was performed using a Leica TCS SP8 MP microscope. For incident light illumination/reflected light widefield recordings (5–10 Hz), we used N PLAN 5x/0.12 or HC APO L10x/0.30 W DLS objectives with large fields of view. Multiphoton time-lapse images were acquired at ~2 Hz frame rates using external hybrid detectors and the HCX IRAPO L25x/0.95 W objective at 930 nm excitation wavelength. Individual recording duration varied between 13 and 30 min (mean = 25 min). For in vivo stimulation experiments, we used a Picospritzer III (Parker Hannifin, Pine Brook, NJ) to puff nanoliter volumes of control saline ($S_1$; containing Alexa

Fluor 555 (4 µM)) or stimulus solution (**S₁**; containing Alexa Fluor 555 (4 µM) and ATP (1 mM)), respectively, from beveled glass micropipettes onto the surface of seminiferous tubules.

## Data analysis

All data were obtained from independent experiments performed on at least three days. Individual numbers of cells/tubules/experiments (n) are denoted in the respective figures and/or legends. If not stated otherwise, results are presented as means ± SEM. Statistical analyses were performed using paired or unpaired *t*-tests, one-way ANOVA with Tukey's HSD *post hoc* test or the Fisher Exact test (as dictated by data distribution and experimental design). Tests and corresponding *p*-values that report statistical significance ($\leq$0.05) are individually specified in the legends. Data were analyzed offline using FitMaster 2.9 (HEKA Elektronik), IGOR Pro 8 (RRID:SCR_000325; WaveMetrics), Excel 2016 (Microsoft, Seattle, WA), and Leica LAS X (RRID:SCR_013673; Leica Microsystems) software. Dose-response curves were fitted by the Hill-equation. Time-lapse live-cell imaging data displaying both $Ca^{2+}$ signals and tubular contractions were analyzed using custom-written code in MATLAB (RRID:SCR_001622; The MathWorks, Natick, MA).

For quantitative image analysis, images from both reflected light and fluorescence time-lapse recordings were registered to their respective first image frame at time point $t_0$, using the registration algorithm from *Liu et al., 2015* (implementation in *Evangelidis, 2013*), resulting in stabilized recordings without movement. For fura-2 fluorescence recordings, we first performed a single registration on the combined image ($f_{340} + f_{380}$) and then applied the displacement vector field, computed by the registration algorithm, to both images ($f_{340}$ and $f_{380}$) separately. ROIs were defined manually at $t_0$ and superimposed onto all subsequent images of the stabilized recording. At each time point $t_i$, the fluorescence signal F was computed as the mean $f_{340}/f_{380}$ ratio of all pixels within a given ROI. When measuring $Ca^{2+}$-dependent changes in GCaMP6f intensity, the fluorescence signal $F$ was normalized with respect to a baseline before stimulation, computing the intensity change for the $i^{th}$ time point as $\frac{F_i - F_{\text{baseline}}}{F_i}$. For clarity, linear baseline shifts were corrected in some example traces.

Seminiferous tubule contractions and transport of luminal content were visualized by reflected light microscopy of acute slices or whole-mount macroscopic tubule imaging, respectively. Data from both types of time-lapse recordings were analyzed and quantified as either flow strength or flow change (see below). For each frame at a given time point $t_i$, the registration algorithm computed a flow or displacement vector field $V_i = \begin{pmatrix} \mathbf{v}_{1,1} & \cdots & \mathbf{v}_{1,n} \\ \vdots & \ddots & \vdots \\ \mathbf{v}_{m,1} & \cdots & \mathbf{v}_{m,n} \end{pmatrix}$, where $\mathbf{v}_{1,1} = (x, y)$ is a vector indicating strength and direction of the displacement of pixel $(1, 1)$ between time points $t_0$ and $t_i$. The average norm $|V_i| = \frac{1}{mn} \sum_{p,q} \|\mathbf{v}_{p,q}\|$ is a measure for the effort that is necessary to register the image at $t_0$ to the image at $t_i$. The flow field strength quantified by this measure is interpreted as the amount of visible changes that, dependent on the experiment, result from tubule contraction and / or luminal content movement. For analysis of contractions in acute seminiferous tubule slices (Figures 3, 4, 7), we quantified the *flow strength* $s_i$ within an ROI as the average norm $|V_i|$ computed only for the $\mathbf{v}_{p,q}$ corresponding to pixels within the ROI defined at $t_0$. For whole-mount macroscopic imaging of luminal content movement in intact tubule segments (*Figure 5*), we quantified the *flow change* $c_i = s_i - s_{i-1}$ as the change of flow strength between two consecutive time points / frames. Here, $s_i$ values were preprocessed by smoothing with a moving average filter. Results are reported as the AUC, that is, the area under the $c_i$ curve.

For analysis of in vivo data, we employed a custom set of ImageJ macros utilizing build-in functions of Fiji-ImageJ (RRID:SCR_002285) (*Rueden et al., 2017*; *Schindelin et al., 2012*). Widefield imaging data was first corrected for brightness fluctuation caused by a 50 Hz AC power supply. Here, we used the *bleach correction* plugin in histogram matching mode (*Miura et al., 2014*). Next, we applied Gaussian filter functions (*GausBlur* (five px radius) and *Gaussian Blur3D* (x = 0, y = 0, z = 5)). We calculated flow change via the *Gaussian Window MSE* function (sigma = 1; max distance = 3). Tubule selection used the polyline tool (line width adjusted to tubule diameter). Selected tubules ranged from 200 µm to 3.4 mm length. Next, flow fields of individual tubules were straightened. Average movement intensity was calculated from transversal line profiles (perpendicular to the

straightened longitudinal axis of each tubule) and plotted as kymographs (space-time plots) to measure movement progression speed from linear regressions.

Multiphoton time-lapse imaging data was recorded in dual-channel mode, with (i) a target channel recording GCaMP6f fluorescence and some background signal (525\50 nm), and (ii) a background channel mainly recording autofluorescence (585\40 nm), allowing for background correction of the GCaMP6f signal using a dye separation routine. Slow constant movement in both channels was registered and removed to correct for steady drift. After Gaussian filtering (*GausBlur* (five px radius); *Gaussian Blur3D* (x = 0, y = 0, z = 5)), flow fields were calculated from the background signal. Again, flow change was calculated via the *Gaussian Window MSE* function (sigma = 1; max distance = 3). Time-lapse epifluorescence in vivo recordings were processed to isolate transient fluorescence signals from static background noise using custom ImageJ code with Fiji's build-in functions (see **Data and materials availability**).

## Acknowledgements

We thank Corinna Engelhardt, Jessica von Bongartz, and Stefanie Kurth (RWTH Aachen University) for assistance, Andreas Meinhardt and Jörg Klug (Justus-Liebig-University Giessen) for kindly providing detailed information about the mouse TPC cell culture protocol, Andrea Mietens and Ralf Middendorff (Justus-Liebig-University Giessen) for comments and suggestions, J Ullrich Schwarzer (Andrology-Center, Munich) and Frank-Michael Köhn (Andrologicum, Munich) for providing human samples, and all members of the Spehr laboratory for discussions.

## Additional information

### Funding

| Funder | Grant reference number | Author |
| --- | --- | --- |
| Deutsche Forschungsgemeinschaft | 368482240/GRK2416 | Marc Spehr<br>Nadine Mundt |
| Deutsche Forschungsgemeinschaft | 412888997 | David Fleck |
| Deutsche Forschungsgemeinschaft | 245169951 | Marc Spehr<br>Artur Mayerhofer |
| Volkswagen Foundation | I/83533 | Marc Spehr |
| Deutsche Forschungsgemeinschaft | 233509121 | Dorit Merhof |

The funders had no role in study design, data collection and interpretation, or the decision to submit the work for publication.

### Author contributions

David Fleck, Conceptualization, Formal analysis, Funding acquisition, Validation, Investigation, Visualization, Methodology, Writing - review and editing; Lina Kenzler, Nadine Mundt, Formal analysis, Validation, Investigation, Visualization, Writing - review and editing; Martin Strauch, Software, Validation, Methodology, Writing - review and editing; Naofumi Uesaka, Investigation, Methodology, Writing - review and editing; Robert Moosmann, Formal analysis, Investigation; Felicitas Bruentgens, Formal analysis, Investigation, Methodology, Writing - review and editing; Annika Missel, Formal analysis, Investigation, Writing - review and editing; Artur Mayerhofer, Supervision, Funding acquisition, Methodology, Writing - review and editing; Dorit Merhof, Supervision, Methodology, Writing - review and editing; Jennifer Spehr, Conceptualization, Formal analysis, Supervision, Validation, Visualization, Methodology, Writing - review and editing; Marc Spehr, Conceptualization, Resources, Data curation, Supervision, Funding acquisition, Visualization, Methodology, Writing - original draft, Project administration

## Author ORCIDs

David Fleck https://orcid.org/0000-0002-6692-2388
Lina Kenzler https://orcid.org/0000-0002-6738-2221
Nadine Mundt https://orcid.org/0000-0003-3370-2933
Robert Moosmann https://orcid.org/0000-0003-2321-7213
Felicitas Bruentgens https://orcid.org/0000-0001-7754-1313
Artur Mayerhofer https://orcid.org/0000-0002-9388-4639
Dorit Merhof https://orcid.org/0000-0002-1672-2185
Marc Spehr https://orcid.org/0000-0001-6616-4196

## Ethics

Animal experimentation: Mice were maintained and sacrificed according to European Union legislation (Directive 2010/63/EU) and recommendations by the Federation of European Laboratory Animal Science Associations (FELASA). All experimental procedures were approved by the State Agency for Nature, Environment and Consumer Protection (LANUV; protocol number / AZ 84-02.04.2016.A371).

## Decision letter and Author response

Decision letter https://doi.org/10.7554/eLife.62885.sa1
Author response https://doi.org/10.7554/eLife.62885.sa2

# Additional files

## Supplementary files

• Transparent reporting form

## Data availability

All data is available in the main text or the supplementary materials. Previously unpublished source code for data analysis (quantification of tubular contractions, flow strength/change, Ca2+ signals) is available at: https://github.com/rwth-lfb/Fleck_Kenzler_et_al Copy archived at https://archive.software-heritage.org/swh:1:rev:88c8792860ddf09fd7da969fef6bf86c40441135/ and https://doi.org/10.5281/zenodo.4280752.

The following dataset was generated:

| Author(s) | Year | Dataset title | Dataset URL | Database and Identifier |
|---|---|---|---|---|
| Fleck D | 2020 | Transient Signal Enhancer for Fiji ImageJ | https://doi.org/10.5281/zenodo.4280752 | Zenodo, 10.5281/zenodo.4280752 |

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
