## [Decision Letter]

**Acceptance summary:**

This paper cuts across the areas of purinergic signaling, smooth muscle control and testis physiology. The authors combine in vitro and well as newly developed elegant in vivo imaging approaches to demonstrate a role for ATP and its receptors in testicular sperm transport. The whole study is very well conducted, the methodology is sound, the statistical analysis is well done, and the authors provide convincing evidence for all their claims. The results are extremely interesting for the fields of reproductive biology and reproductive medicine, and the whole field of physiology.

**Decision letter after peer review:**

Thank you for submitting your article "ATP activation of peritubular cells drives testicular sperm transport" for consideration by *eLife*. Your article has been reviewed by three peer reviewers, and the evaluation has been overseen Anna Akhmanova as the Senior and Reviewing Editor. The following individuals involved in review of your submission have agreed to reveal their identity: Ivan Manzini (Reviewer #1); Johannes Reisert (Reviewer #3).

The reviewers have discussed the reviews with one another and the Reviewing Editor has drafted this decision to help you prepare a revised submission.

All three reviewers agreed that this is an excellent paper that adds a lot of new information to the field. However, the reviewers also agreed that to make this paper suitable for the broad readership of *eLife*, it is essential to revise and extend both the Introduction and the Discussion and to improve data presentation according to the specific comments listed below. Furthermore, reviewer 3 would particularly like to see clarification of the roles of P2X vs. P2Y receptors and also the potential dichotomy of the spread of contraction vs. the spread of Ca signaling.

Reviewer #1:

The manuscript entitled "ATP activation of peritubular cells drives testicular sperm transport" by Davis Fleck and co-workers deals with the involvement of purinergic signaling in testicular sperm transport.

To obtain their results the authors used several state-of-the-art methods. In many of the conducted experiments they skilfully combined some of these methods to obtain unprecedented results. The authors verified their results employing different testis preparations (from cultured cells to in vivo recordings) and using different calcium sensors. They developed a very elegant custom-built 3D printed imaging stage that allowed experiments in the testis of living mice.

The whole study is very well conducted, the methodology is sound, the statistical analysis is well done, and the authors provide convincing evidence for all their claims. They unambiguously show that nucleotide-triggered coordinated contractions of smooth muscle-like testicular peritubular cells provide the propulsive force for luminal sperm cell transport along seminiferous tubules.

The results are extremely interesting for the fields of reproductive biology and reproductive medicine, but also for the whole field of physiology. The obtained results will with no doubt significantly influence the future research in the relevant fields.

I do feel, however, that prior to publication a few points (see below) need to be clarified. Clarifying this points will improve the comprehensibility and readability of the manuscript.

1) Abstract: The authors mention peritubular Ca^2+^ waves in the Abstract, but do not mention them again in the whole manuscript. This is a little disturbing. I expected to find more information about these waves throughout the manuscript. How do the authors define these waves? Please clarify.

2) Although I generally like concise introductions, I feel that in the present manuscript the Introduction is too short. The authors should give more information about the cellular composition of the seminiferous tubules, the stages of sperm cell maturation, and the current knowledge of sperm cell transport in the tubules. The authors should consider including a schematic figure that gives an easy to understand overview of the above points. This will make all the performed experiments and the whole story easier to understand, especially for readers that are not experts in the field of reproductive biology and medicine.

3) Subsection “ATP triggers seminiferous tubule contractions” and corresponding Figure 4E-G: I have difficulties to understand what is shown here. It might be only me, but the authors should consider trying to explain this part of their results better.

4) Subsection “ATP drives directional luminal transport”: The authors state that in their preparation it is possible to visually categorize the twelve seminiferous epithelium cycle stages. I have difficulties to identify them. The authors should consider giving more information what exactly this twelve cycle stages are and how they can be distinguished in the picture (corresponding Figure 5). Part of this information could be included in an overview figure (see my first point above).

5) Subsection “ATP drives directional luminal transport”: In this subsection the authors state that their finding indicates that ATP acts as a local paracrine messenger that, by itself, is not sufficient to trigger a signal that propagates in a wave-like fashion along the tubules. They speculate that transport directionality in the tubules likely results from morphological characteristics rather than peristaltic contractility.

What do the authors mean with "morphological characteristics" and how can these characteristics explain the transport directionality? Also, why is the transport directionality probably not connected to peristaltic contractility?

6) Subsection “ATP induces tubular contractions in vivo”: In this subsection the authors state that the movement analysis along the length of digitally straightened tubules demonstrates wave-like unidirectional motions that propagate with high velocities. What factors do drive this propagation? The authors should consider adding some information.

7) Discussion: I generally like concise discussions, but in my opinion in this manuscript also the Discussion, like the Introduction, is way too short. It is more like a short conclusion. Here a few points that could/ should be speculated about:

– What could be the in vivo origin of ATP (cell type, production site)?

– What could be the trigger for the physiological ATP release?

– Why do nucleotides act "only" as paracrine mediators and do not induce calcium waves like in other tissues?

– The authors could speculate about how likely it is that the same mechanisms are present also in human testis.

– The authors could speculate about how their results could help finding treatments for infertility and / or be useful for the development of male contraception.

– etc.

8) I had problems with Videos 5 and 8. Somehow they did not run on my computers. I know that.mov-files are often requested, but there are certainly better formats for online videos. It could be a problem that my computers have, but I recommend checking these videos. Videos in an MP4 format would probably be better.

Reviewer #2:

This manuscript by Fleck et al. investigates the molecular mechanism regulating smooth muscle contractions in the testis. Overall, this is an excellent manuscript using innovative imaging techniques to visualize almost simultaneously intracellular Ca^2+^ and peritubular cells movement. In addition, conclusions throughout the manuscript are sound and figure legends are clear. Because this manuscript is relevant for different fields, the authors should improve the writing significantly. First, the Introduction is very superficial and does not elaborate on the rationale behind this manuscript. Moreover, the authors assume that reproductive biologists are familiar with purinergic signaling and avoid any explanation on the pathways. Although short introductions are usually welcome, this one in particular has only two lines more than the Abstract (21 vs. 19 lines). The Results section is better; however, there are many sections in which the authors give for granted knowledge on the system (e.g. the use of specific agonists and antagonists for purinergic receptors; depletion of intracellular Ca^2+^ stores; and others). Finally, if the Introduction was short, the Discussion is almost absent (only 15 lines). The Discussion is also superficial, repeating some sections from the Abstract and Introduction. As the authors said, the topic of smooth muscle regulation in the testis has been underexplored for many years, there is much to say on the relevance of their findings that cannot be written in such a short Discussion.

1) Abstract: "Here, we show that coordinated contractions of smooth muscle-like testicular peritubular cells provide the propulsive force for luminal sperm transport toward the rete testis and epididymis." This sentence should be toned down; I agree that this manuscript shows coordinated contractions of smooth muscle. This finding suggests a possible mechanism for the propulsive force required for sperm transport; however, the manuscript does not make a direct observation of the sperm movement through the rete testis and epididymis.

2) In the Abstract, the authors claim "Consequently, our findings could have substantial pharmaceutical implications for both infertility treatment and / or male contraception." As the authors recognized at the end of the manuscript, this sentence is speculative and does not belong to the Abstract.

3) Introduction: The Introduction is not informative. The authors should give information on the purinergic system and its role in regulating smooth muscle. They should also incorporate to the Introduction text that let the reader knows on the main signaling pathways involved in purinergic signaling as well as on the type of purinergic receptors, agonists and antagonists.

4) Subsection “ATP is a potent TPC stimulus”, explain briefly the conclusion that P2X2 or P2X4 but not P2X7 are responsible for the currents observed.

5) Subsection “ATP is a potent TPC stimulus”: Explain the rationale of the conclusion "this current likely results from P2Y receptor-mediated phosphoinositide turnover" Why? Also, the authors state that these cells are sensitive to UTP and dependent on the presence of intracellular GTP. How were these experiments designed? Which is the conclusion of these findings? In general, more rationale is needed to explain why the experiments were conducted. For example, in Figure 1, ivermectin and suramin are used without introducing the mode of action of these compounds.

6) Subsection “ATP is a potent TPC stimulus”: Explain briefly how the Ca^2+^ was reduced to 100 nM to avoid the reader to go to Materials and methods while reading the Results.

7) Subsection “ATP is a potent TPC stimulus”: The UTP action is mentioned; however, only later in the same paragraph, the authors state that UTP is a selective P2Y agonist.

Overall, the results are clear; however, it is very difficult to follow the rationale from the Results section text. As written this section is not more helpful than the figure legend.

8) Subsection “ATP is a potent TPC stimulus”. Give a rationale for the use of GCaMP6f, explain briefly the strategy.

9) Subsection “ATP is a potent TPC stimulus”: Define Ai14D mice, define SMMHC.

Subsection “ATP triggers seminiferous tubule contractions”: Briefly explain the high frequency rippling observed.

10) Subsection “ATP triggers seminiferous tubule contractions”: Mention the pharmacological inhibitor used and the time used for the treatment to avoid the reader to go to the Materials and methods section. In Figure 4 legend, the authors mentioned that they use CPA (please define) and they state that they pre-incubate for >= 15 min., could the time frame be more precise?

11) Subsection “ATP triggers seminiferous tubule contractions”: Conclusions 'a' to 'c' are clear. Regarding the last conclusion 'd', briefly re-state the evidence supporting it; without stating the evidence, this conclusion is not sound.

12) Figure 4. This figure lacks explanation in the main text. The reader needs to carefully analyze Figure 4 legend to understand the logic of the experiments performed. A summary of each panel in the main text will be helpful and should be added. In particular, there is no explanation of panel H in the main text. This panel is difficult to follow even after careful reading of the figure legend. The authors should make a significant effort to explain their thought process to scientists that are not necessarily experts in the field of purinergic signaling.

13) Discussion:

The authors speculate that their findings on purinergic signaling in the testis can have implications for male infertility and/or contraception. This sentence is repeated from the Abstract. Although there is nothing wrong to mention some putative translational implication, as written this sentence is not connected with the rest of the manuscript. To leave this sentence in a final manuscript, the authors should elaborate a rationale on how these findings can be used for either improving male fertility or for contraception. On the other hand, this speculation is not an important part of the manuscript and does not belong to the Abstract.

14) Overall, the Discussion section is short and superficial. There is almost no analyses of the findings; some of the questions that should be addressed are: How results in this manuscript relates to other works in the area? Which are the unknowns that will need further investigation (gap in the knowledge)? How the purinergic system in the testis can be related to clinical applications (male fertility and/or contraception)? Which are the advances obtained in terms of imaging methodologies? How the authors think that the system is regulated: is it paracrine as they propose? If this is the case, what would be the signaling for the release of ATP? It is ok not to know the answer, it is not ok to avoid discussing these topics.

15) In summary, this is an excellent manuscript; however, it is written as a lab report for internal consumption. In general, the Results section needs a little more explanation on the rationale behind the experimental strategy and some clarification on the results. On the other hand, the authors should make a significant effort to improve the Introduction and Discussion section.

Reviewer #3:

The manuscript by Fleck and colleagues addresses the role of ATP and its receptors in sperm transport in the seminiferous tubules in the testis. They conclude that ATP, in a paracrine manner, drives intracellular Ca^2+^ signals in testicular peritubular cells (TPCs) and hence their contractions in a directional manner. The manuscript is mostly very clearly laid out, experiments are well designed using broad and also entirely novel and quite impressive methodologies for in vivo recordings. Their results are an exciting and important contribution to a field fraught with confusion and they will lead to a deeper and further understanding of transport in seminiferous tubules.

1) The roles of P2X and P2Y receptors and their contribution to Ca^2+^ signaling in TPCs is somewhat confusing. In Figure 1J, the authors demonstrate that the Ca^2+^ signal in TPCs originates both from extracellular sources (P2X mediated) as well as intracellular sources (P2Y mediated). But depletion of intracellular stores (Figure 4D) seems to entirely eliminate the cellular Ca^2+^ signal, suggesting P2Y mediated signaling only? Also, given that P2Y-mediated Ca^2+^ signals do play a role, it might be worth considering moving data in Figure 1—figure supplement 1 into Figure 1 of the main paper to give them more prominence.

2) Figures 5 and 6 describe directional luminal transport and TPC Ca^2+^ signals respectively following focal application of ATP, with contraction occurring in a stage-dependent directional manner, while the Ca^2+^ signal is confined locally to where ATP was applied with no spread. This seems puzzling as it raises the question of what signal drives the contractions (and their directionality) in the TPCs that contract (one assumes) without an apparent Ca^2+^ signal?

3) Along similar lines, what signal drives the directionality of the contractions? ATP diffusing along the tubules? Or Ca^2+^ waves spreading from cell to cell (although that should become evident in the Ca^2+^ imaging)? While this might be outside the scope of the manuscript, it might be worth addressing this issue in the currently quite focused Discussion.

---

## [Author Response]

Reviewer #1:[…] The results are extremely interesting for the fields of reproductive biology and reproductive medicine, but also for the whole field of physiology. The obtained results will with no doubt significantly influence the future research in the relevant fields.I do feel, however, that prior to publication a few points (see below) need to be clarified. Clarifying these points will improve the comprehensibility and readability of the manuscript.1) Abstract: The authors mention peritubular Ca^2+^ waves in the Abstract, but do not mention them again in the whole manuscript. This is a little disturbing. I expected to find more information about these waves throughout the manuscript. How do the authors define these waves? Please clarify.

Reviewer 1 makes a valid point. We failed to refer to the “Ca^2+^ waves” in the original manuscript (aside from the Abstract). This has now been corrected. We now show a representative Ca^2+^ wave that travels in a unidirectional manner along a tubule’s longitudinal axis (new Figure 7—figure supplement 2D). We use a sequence of successive single frames from time-lapse in vivo recordings (5 Hz image acquisition rate), in which we overlay a brightfield image with a pseudocolor fluorescence intensity image that indicates elevations in TPC Ca^2+^ concentration. The wave-like character also becomes apparent from an overlay of traces (fluorescence intensity over time) corresponding to adjacent regions-of-interest in the same tubule.

Accordingly, we have now revised the Results and Discussion sections, which now read:

Results: “…Movement analysis along the length of digitally straightened tubules demonstrates wave-like unidirectional motions that propagate with high velocities (Figure 7C and D). These movements coincide with ‘macroscopic’ Ca^2+^ waves that travel at comparable speed and direction (Figure 7—figure supplement 2D). Notably, the observed coordinated contractile movements…”

Results: “…Taken together, in vivo recordings demonstrate that robust recurrent seminiferous tubule contractions (i) occur spontaneously, (ii) are driven by cytosolic Ca^2+^ elevations in TPCs that propagate in a wave-like fashion, and (iii) can be triggered experimentally by ATP exposure.…”

Discussion: “…While peristaltic contractions are driven by propagating wave-like Ca^2+^ signals in vivo, focal ATP stimulation appears insufficient to trigger a regenerative Ca^2+^ wave in isolated tubules. We, thus, conclude…”

The legend of Figure 7—figure supplement 2 has been revised accordingly.

The custom code used for image processing is made available by link in the **“**Data and materials availability” subsection.

2) Although I generally like concise introductions, I feel that in the present manuscript the Introduction is too short. The authors should give more information about the cellular composition of the seminiferous tubules, the stages of sperm cell maturation, and the current knowledge of sperm cell transport in the tubules. The authors should consider including a schematic figure that gives an easy to understand overview of the above points. This will make all the performed experiments and the whole story easier to understand, especially for readers that are not experts in the field of reproductive biology and medicine.

We thank reviewer 1 for raising this important point and for his insightful suggestions. We acknowledge his argument and, accordingly, we have now added “a schematic figure that gives an easy to understand overview of the above points”. This new schematic diagram is now introduced as Figure 1A and it illustrates both “cellular composition of the seminiferous tubules” and “the stages of sperm cell maturation.” The legend of Figure 1 has been revised accordingly.

We have also added new text to the Introduction, which now reads: “The seminiferous epithelium is composed of Sertoli cells, each intimately associated with ≥30 germ cells at different developmental stages (Mruk and Cheng, 2004). […] After approximately 39 days (4.5 cycle repetitions), spermatogenesis completes with the release of immotile spermatozoa from the seminiferous epithelium into the lumen of the tubule (spermiation).”

3) Subsection “ATP triggers seminiferous tubule contractions” and corresponding Figure 4E-G: I have difficulties to understand what is shown here. It might be only me, but the authors should consider trying to explain this part of their results better.

Reviewer 1 emphasizes an important point. We apologize for the somewhat vague description of our approach and, accordingly, we have extensively revised the corresponding Results section.

We added several explanatory remarks to the relevant paragraphs, which now read: “First, we asked whether influx of external Ca^2+^ is involved in TPC force generation. […] Together, these data strongly suggest that (i) extracellular ATP acts as a potent TPC stimulus that triggers seminiferous tubule contractions in situ, that (ii) P2X and P2Y receptors act in concert to mediate TPC responses to ATP exposure, that (iii), while P2X receptor-dependent external Ca^2+^ influx apparently boosts responses to ATP, P2Y receptor-mediated Ca^2+^ mobilization from the sarcoplasmic reticulum is necessary to evoke TPC responses, and consequently that (iv) influx of external Ca^2+^ via ionotropic P2X receptors is not sufficient to drive TPC signals and evoke contractions. Notably, our general finding…”

In addition, we added explanatory notes to the legend of Figure 4, which now reads: “(F) Signal amplitudes (Ca^2+^, black; contractions, red) of responding TPCs / tubules, quantified as a function of stimulus, treatment, and Ca^2+^ sensor. Data (mean ± SEM) are normalized to the respective initial responses to ATP (10 µM) under control conditions (dotted horizontal line; see first stimulations in (B and C)).”

4) Subsection “ATP drives directional luminal transport”: The authors state that in their preparation it is possible to visually categorize the twelve seminiferous epithelium cycle stages. I have difficulties to identify them. The authors should consider giving more information what exactly this twelve cycle stages are and how they can be distinguished in the picture (corresponding Figure 5). Part of this information could be included in an overview figure (see my first point above).

We thank reviewer 1 for pointing this out. Our wording has obviously been misleading. In fact, we cannot identify each of the twelve seminiferous epithelium cycle stages in the whole-mount preparation that we established. What we can do, however, is following the approach first proposed by Hess and De Franca, 2008, and categorize stages I-V as ‘early’, stages VI-VIII as ‘middle’, and stages IX-XII as ‘late’. The corresponding ‘stage groups I, II and III’ are characterized by different densities of luminal sperm that can be distinguished clearly by standard widefield microscopy.

To clarify, we followed the expert’s advice and have now added this information to “an overview figure” (Figure 1A; also see response to (2) above). We now also refer to the three groups again in Figure 5A, using the same color code. Moreover, we have revised the Results that now reads:

“In addition, this setup enables visual categorization of the spermatogenic cycle into three distinct stage groups following published protocols (Hess and De Franca, 2008) and allows precisely timed focal perfusion (Materials and methods). […] When we analyzed ATP-induced movement in directly stimulated regions (each designated as region-of-interest (ROI) 0), comparing stage groups with a high (group II) versus a relatively low (groups I and III) amount of luminal sperm, we observed no difference in stimulation-dependent motion (Figure 5C_I_).”

5) Subsection “ATP drives directional luminal transport”: In this subsection the authors state that their finding indicates that ATP acts as a local paracrine messenger that, by itself, is not sufficient to trigger a signal that propagates in a wave-like fashion along the tubules. They speculate that transport directionality in the tubules likely results from morphological characteristics rather than peristaltic contractility.What do the authors mean with "morphological characteristics" and how can these characteristics explain the transport directionality? Also, why is the transport directionality probably not connected to peristaltic contractility?

Reviewer 1 raises another important point that is also brought up by reviewer 3 (main points 2 and 3). It is thus becoming clear that we did not manage to communicate these results and, specifically, our interpretation in a clear and compelling way. We apologize and, accordingly, we have extensively revised both the corresponding Results text and the Discussion.

In Figure 5, we show that focal application of ATP triggers contractions that translate into movement of luminal content. This read-out (luminal flow change 𝑐𝑐_𝑖𝑖_) is different from the contraction measurements (flow strength 𝑠𝑠_𝑖𝑖_) conducted in Figures 3, 4 and 7. We describe the difference in the “Data analysis” subsection of the Materials and methods. We believe that this important distinction did not become sufficiently clear in the previous version of the manuscript. In a nutshell, the results shown in Figure 5 cannot be interpreted as evidence for propagating longitudinal contractions (nor do they argue against such a phenomenon). What the results *do* show is that (i) a local ATP stimulus provides the impulse to push luminal content – with or without triggering a peristaltic wave – and that (ii) for epithelial regions in stage group II, this movement displays directionality.

In Figure 6, we then ask if the underlying mechanism might be a peristaltic contraction, which we do assume must be causally related to TPC Ca^2+^ signals. Since we do not find evidence for a regenerative Ca^2+^ wave (Figure 6), we assume that, under the present experimental conditions, other features are involved. “Morphological characteristics” such as an anatomical setting in which Sertoli cells effectively function as ‘nonreturn valves’ could, for example, be involved (see below).

The absence of a regenerative Ca^2+^ wave in isolated seminiferous tubules could, of course, have several reasons, including any number of missing environmental factors present in the intact testis or simply the inability to register low-amplitude Ca^2+^ signals when using large field-of-view / low numerical aperture objectives. Importantly, we always detect coincident Ca^2+^ signals when monitoring seminiferous tubule contractions in vivo (Figure 7A and B, Video 6). Moreover, propagation of wave-like Ca^2+^ signals is also regularly observed in vivo (new Figure 7—figure supplement 2D).

In the revised manuscript, we now make an effort to describe the data and their interpretation in clear terms in the Results. Moreover, we have added a dedicated paragraph to the Discussion that picks up on the issue of Ca^2+^ signal – contraction coupling and propagation. Specifically, we have revised / added the following text:

Results: “Third, we investigated if luminal movement is restricted to the area of stimulation or, by contrast, if fluid flow propagates beyond the directly stimulated tubule section. […] These findings demonstrate directionality of sperm transport upon focal purinergic TPC stimulation in isolated seminiferous tubules.”

Results: “As expected, ATP-induced tubule contractions also manifest as Ca^2+^ signals in TPCs (Figure 6A and B, Video 5). […] However, local contractions generate sufficient force to move luminal content beyond the directly stimulated area and, in turn, directionality of flow along short-to-medium distances (≤600 µm; Figure 5C_II_) is not critically dependent on peristaltic contractility.”

Discussion: “Whole-mount imaging of isolated seminiferous tubules reveals propagation of luminal content that extends beyond the confines of the stimulated / contracted area and displays stage-dependent directionality. […] We cannot, however, rule out that the use of large field-of-view / low numerical aperture objectives for ‘macroscopic’ imaging simply prevents the detection of low-amplitude Ca^2+^ signal spread.”

Discussion: “Among several remaining questions, future experimental efforts will have to address (i) whether TPCs are coupled by gap junctions to display coordinated activity; (ii) whether… […] …(viii) whether an additional cytosolic and / or membrane Ca^2+^ oscillator (Berridge, 2008) provides an endogenous pacemaker mechanism that acts independent of purinergic stimulation; and…”

6) Subsection “ATP induces tubular contractions in vivo”: In this subsection the authors state that the movement analysis along the length of digitally straightened tubules demonstrates wave-like unidirectional motions that propagate with high velocities. What factors do drive this propagation? The authors should consider adding some information.

Again, reviewer 1 makes a valid point that also refers to his points 1, 5, and 7 (see above and below, respectively). As outlined in our response to these points, the primary factor that drives this propagation is wave-like TPC Ca^2+^ elevations, which we now refer to several times in the manuscript and we also show a representative example (new Figure 7—figure supplement 2D). The detailed mechanisms underlying the spread of the Ca^2+^ signal and, therefore, the propagation of contractions / luminal content motion currently remain subject to debate and speculation. Accordingly, we have now added several relevant new paragraphs to the Discussion (see our response to point 7 below).

7) Discussion: I generally like concise discussions, but in my opinion in this manuscript also the Discussion, like the Introduction, is way too short. It is more like a short conclusion.

The reviewer makes an excellent suggestion. We acknowledge his argument and, therefore, we have now substantially revised the Discussion. Several new paragraphs have been added to address the reviewers concerns.

Specifically, the following text has been added.

Here a few points that could/ should be speculated about:– What could be the in vivo origin of ATP (cell type, production site)?

Discussion: “The site(s) / cellular origin of testicular ATP release as well as the mechanism(s) that trigger ATP secretion in vivo currently remain elusive. […] Thus, TPCs could themselves participate in regenerative nucleotide release.”

– What could be the trigger for the physiological ATP release?

Discussion: “During spermatogenesis, apoptosis is a vital process (Print and Loveland, 2000). […] In addition, ATP release has been observed in several cell types as a result of mechanical deformation, shear stress, stretch, or osmotic swelling (Button et al., 2013) adding another putative mechanism of regenerative signaling in purinergic contraction control.”

– Why do nucleotides act "only" as paracrine mediators and do not induce calcium waves like in other tissues?

We have now addressed this point in the Discussion. For brevity, we refer to our response to concern 5 (see above).

– The authors could speculate about how likely it is that the same mechanisms are present also in human testis.

Discussion: “Translation of our findings from the mouse model to humans awaits further in-depth investigation. […] While a single layer of TPCs surrounds the mouse seminiferous tubules, the human tubular wall architecture is more complex, containing several TPC layers, substantial amounts of extracellular matrix proteins, and immune cells (Mayerhofer, 2013).”

– The authors could speculate about how their results could help finding treatments for infertility and / or be useful for the development of male contraception.

Discussion: “Impaired spermatogenesis in sub- / infertile men typically coincides with tubular wall remodeling and a partial loss of TPC contractility proteins has been reported in infertile men (Welter et al., 2013). […] Accordingly, pharmacological targeting of purinergic signaling pathways to (re)gain control of TPC contractility represents an attractive approach for male infertility treatment or contraceptive development.”

8) I had problems with Videos 5 and 8. Somehow they did not run on my computers. I know that.mov-files are often requested, but there are certainly better formats for online videos. It could be a problem that my computers have, but I recommend checking these videos. Videos in an MP4 format would probably be better.

We acknowledge the reviewer’s point. We have checked the.mov files on several PCs and found no difficulties. Yet, we have also rendered the video files as.mp4 and can provide them upon request should the editors prefer this format, which produces somewhat larger files.

Reviewer #2:This manuscript by Fleck et al. investigates the molecular mechanism regulating smooth muscle contractions in the testis. Overall, this is an excellent manuscript using innovative imaging techniques to visualize almost simultaneously intracellular Ca^2+^ and peritubular cells movement. In addition, conclusions throughout the manuscript are sound and figure legends are clear. Because this manuscript is relevant for different fields, the authors should improve the writing significantly. First, the Introduction is very superficial and does not elaborate on the rationale behind this manuscript. Moreover, the authors assume that reproductive biologists are familiar with purinergic signaling and avoid any explanation on the pathways. Although short introductions are usually welcome, this one in particular has only two lines more than the Abstract (21 vs. 19 lines). The Results section is better; however, there are many sections in which the authors give for granted knowledge on the system (e.g. the use of specific agonists and antagonists for purinergic receptors; depletion of intracellular Ca^2+^ stores; and others). Finally, if the Introduction was short, the Discussion is almost absent (only 15 lines). The Discussion is also superficial, repeating some sections from the Abstract and Introduction. As the authors said, the topic of smooth muscle regulation in the testis has been underexplored for many years, there is much to say on the relevance of their findings that cannot be written in such a short Discussion.1) Abstract: "Here, we show that coordinated contractions of smooth muscle-like testicular peritubular cells provide the propulsive force for luminal sperm transport towards the rete testis and epididymis." This sentence should be toned down; I agree that this manuscript shows coordinated contractions of smooth muscle. This finding suggests a possible mechanism for the propulsive force required for sperm transport; however, the manuscript does not make a direct observation of the sperm movement through the rete testis and epididymis.

Reviewer 2 makes a valid point. We did not intend to claim that we “make a direct observation of the sperm movement through the rete testis and epididymis.” In fact, we are not able to make any statement that refers to the passage of sperm within the rete testis and epididymis. We would like to respectfully point out to the reviewer that we intentionally referred to “sperm transport *towards* the rete testis”, not “*through* the rete testis.” We believe that this is still a valid point to make. In conclusion, we acknowledge the reviewer’s argument and, accordingly, we have deleted the reference to the epididymis and rephrased the relevant sentence, which now reads:

“Here, we show that coordinated contractions of smooth muscle-like testicular peritubular cells provide the propulsive force for luminal sperm transport towards the rete testis.”

2) In the Abstract, the authors claim "Consequently, our findings could have substantial pharmaceutical implications for both infertility treatment and / or male contraception." As the authors recognized at the end of the manuscript, this sentence is speculative and does not belong to the Abstract.

We acknowledge the reviewer’s argument and, accordingly, we have now deleted this speculative sentence from the Abstract.

3) Introduction: The Introduction is not informative. The authors should give information on the purinergic system and its role in regulating smooth muscle. They should also incorporate to the Introduction text that let the reader knows on the main signaling pathways involved in purinergic signaling as well as on the type of purinergic receptors, agonists and antagonists.

Reviewer 2 emphasizes an important point. We agree that the Introduction, in its original form, may have been too superficial. Thus, we acknowledge the reviewer’s argument and, accordingly, we have now added “information on the purinergic system and its role in regulating smooth muscle” as well as on “the main signaling pathways involved in purinergic signaling”. The relevant paragraphs in the revised Introduction now read:

“Accumulating evidence implicates purinergic signaling in testicular paracrine communication. […] Functionally, several studies have suggested purinergic paracrine control of gonadotropin effects on Leydig and Sertoli cells (Filippini et al., 1994; Gelain et al., 2005, 2003; Lalevée et al., 1999; Meroni et al., 1998; Poletto Chaves et al., 2006), including steroidogenesis and testosterone / 17β-estradiol secretion (Foresta et al., 1996; Rossato et al., 2001).”

“Members of the P2 purinoceptor family are activated by extracellular ATP (Burnstock, 1990). […] So far, the most prominent role for a specific subunit in reproductive physiology has been attributed to P2X1, which is critical for vas deferens smooth muscle contraction and male fertility (Mulryan et al., 2000).”

“In mice, stimulation-dependent ATP secretion from both Sertoli and germ cells was reported (Gelain et al., 2005, 2003) and may itself be under endocrine control (Gelain et al., 2005; Lalevée et al., 1999). […] Alternative ATP release pathways include connexin / pannexin hemichannels (Bao et al., 2004; Cotrina et al., 1998), transporters (Lohman et al., 2012), voltage-gated (Taruno et al., 2013) or large-conductance anion (Bell et al., 2003) channels, or even P2X7 receptors (Pellegatti et al., 2005; Suadicani et al., 2006).”

4) Subsection “ATP is a potent TPC stimulus”, explain briefly the conclusion that P2X2 or P2X4 but not P2X7 are responsible for the currents observed.

We have added an explanatory section to the relevant Results paragraph, which now reads:

“The specific biophysical and pharmacological profile of ATP-dependent transmembrane currents (Figure 1D–H) strongly suggests functional expression of P2X2 and / or P2X4, but not P2X7 receptors. […] Ivermectin (Figure 1G and H), an agent selectively potentiating P2X4 receptor currents (Khakh et al., 1999; Silberberg et al., 2007), increased ATP-induced currents in a subpopulation of TPCs (n = 7/12), whereas suramin (Figure 1G and H), a drug inhibiting P2X2, but not P2X4 receptors (Evans et al., 1995), inhibited a TPC subset (n = 10/18).”

5) Subsection “ATP is a potent TPC stimulus”: Explain the rationale of the conclusion "this current likely results from P2Y receptor-mediated phosphoinositide turnover" Why? Also, the authors state that these cells are sensitive to UTP and dependent on the presence of intracellular GTP. How were these experiments designed? Which is the conclusion of these findings? In general, more rationale is needed to explain why the experiments were conducted. For example, in Figure 1, ivermectin and suramin are used without introducing the mode of action of these compounds.

We thank reviewer 2 for pointing this out. We acknowledge his / her argument and, accordingly, we have extensively revised the corresponding Results paragraph, which now reads:

“Notably, live-cell ratiometric Ca^2+^ imaging in cultured TPCs revealed robust and repetitive cytosolic Ca^2+^ transients upon ATP exposure (Figure 1I and J). […] Together, these data suggest that mouse TPCs functionally express both ionotropic and metabotropic purinoceptors.”

The modes of action of “ivermectin and suramin” have now also been described (please see our response to point 4 above).

6) Subsection “ATP is a potent TPC stimulus”: Explain briefly how the Ca^2+^ was reduced to 100 nM to avoid the reader to go to Materials and methods while reading the Results.

This has been done. The relevant sentence now reads: “We next reduced the extracellular Ca^2+^ concentration ([Ca^2+^]_e_) to 100 nM, a concentration approximately equimolar to cytosolic levels, by adding an appropriate chelator / ion concentration ratio (1 mM EGTA / 0.5 mM CaCl_2_). This treatment, which drastically diminishes the driving force for Ca^2+^ influx, did substantially reduce, but not abolish ATP response amplitudes (Figure 1K and L).”

7) Subsection “ATP is a potent TPC stimulus”: The UTP action is mentioned; however, only later in the same paragraph, the authors state that UTP is a selective P2Y agonist.

This has now been corrected (please see our response to point 5 above).

8) Subsection “ATP is a potent TPC stimulus”. Give a rationale for the use of GCaMP6f, explain briefly the strategy.

This has now been done. The relevant sentence now reads:

“In parallel approaches, we employed two different fluorescent Ca^2+^ reporters, a synthetic ratiometric Ca^2+^ sensor (fura-2) as well as a genetically encoded Ca^2+^ indicator (GCaMP6f). […] By contrast, conditional gene targeting via the Cre/Lox system

(Smith, 2011) allows TPC-specific expression of the single-wavelength indicator GCaMP6f.”

9) Subsection “ATP is a potent TPC stimulus”: Define Ai14D mice, define SMMHC.

This has been done. The relevant sentence now reads:

“Tamoxifen-induced transgenic expression of CreER^T2^ under control of the mouse smooth muscle myosin, heavy polypeptide 11 (a.k.a. SMMHC) promoter drives Cre-mediated recombination of loxP-flanked reporters (tdTomato (Ai14D mice) or GCaMP6f (Ai95D)) in smooth muscle cells and TPCs (Wirth et al., 2008).”

10) Subsection “ATP triggers seminiferous tubule contractions”: Mention the pharmacological inhibitor used and the time used for the treatment to avoid the reader to go to the Materials and methods section. In Figure 4 legend, the authors mentioned that they use CPA (please define) and they state that they pre-incubate for >= 15 min., could the time frame be more precise?

Cyclopiazonic-acid (CPA), a blocker of the sarco / endoplasmic reticulum Ca^2+^-ATPase, is now defined (subsection “ATP triggers seminiferous tubule contractions”).

Regarding the duration of CPA treatment, we now added “more precise” information to the legend of Figure 4D (i.e., 18.8 ± 9.3 min). To explain the variability in incubation time, we now also added an explanatory paragraph to the “Fluorescence Ca^2+^ imaging” section in Materials and methods, which reads:

“To ensure effective store depletion by CPA treatment, we recorded intracellular Ca^2+^ levels during CPA incubation at low frequency to monitor Ca^2+^ release from the ER, but also prevent phototoxicity. […] The time-course of this Ca^2+^ release – Ca^2+^ extrusion process varied between samples and ranged between 5.3 and 44.0 min (18.8 ± 9.3 min; mean ± SD).”

11) Subsection “ATP triggers seminiferous tubule contractions”: Conclusions 'a' to 'c' are clear. Regarding the last conclusion 'd', briefly re-state the evidence supporting it; without stating the evidence, this conclusion is not sound.

We thank reviewer 2 for raising this concern. We acknowledge his/her argument and, accordingly, we have revised the relevant Results paragraph that now reads:

“Together, these data strongly suggest that (i) extracellular ATP acts as a potent TPC stimulus that triggers seminiferous tubule contractions in situ, that (ii) P2X and P2Y receptors act in concert to mediate TPC responses to ATP exposure, that (iii), while P2X receptor-dependent external Ca^2+^ influx apparently boosts responses to ATP, P2Y receptor-mediated Ca^2+^ mobilization from the sarcoplasmic reticulum is necessary to evoke TPC responses, and consequently – since store depletion essentially abolishes ATP-dependent signals – that (iv) influx of external Ca^2+^ via ionotropic P2X receptors is not sufficient to drive TPC signals and evoke contractions.”

12) Figure 4. This figure lacks explanation in the main text. The reader needs to carefully analyze Figure 4 legend to understand the logic of the experiments performed. A summary of each panel in the main text will be helpful and should be added. In particular, there is no explanation of panel H in the main text. This panel is difficult to follow even after careful reading of the figure legend. The authors should make a significant effort to explain their thought process to scientists that are not necessarily experts in the field of purinergic signaling.

Reviewer 2 makes a valid point. Accordingly, we have now made “a significant effort to explain” the rationale, particularly with respect to Figure 4H. The substantially revised Results paragraph that now reads:

“We next investigated the Ca^2+^ signaling mechanism(s) underlying ATP-dependent TPC contractions. […] Notably, these UTP responses were statistically indistinguishable from the diminished ATP-dependent signals we observed under low [Ca^2+^]_e_ conditions (Figure 4F).”

13) Discussion:The authors speculate that their findings on purinergic signaling in the testis can have implications for male infertility and/or contraception. This sentence is repeated from the Abstract. Although there is nothing wrong to mention some putative translational implication, as written this sentence is not connected with the rest of the manuscript. To leave this sentence in a final manuscript, the authors should elaborate a rationale on how these findings can be used for either improving male fertility or for contraception. On the other hand, this speculation is not an important part of the manuscript and does not belong to the Abstract.

We agree and thank reviewer 2 for pointing this out. As mentioned above (see response to point 2) we deleted this sentence from the Abstract. In agreement with a request from reviewer 1 (point 7), we have now added “a rationale on how these findings can be used for either improving male fertility or for contraception” to the Discussion, which now reads:

“Impaired spermatogenesis in sub- / infertile men typically coincides with tubular wall remodeling and a partial loss of TPC contractility proteins has been reported in infertile men (Welter et al., 2013). […] Accordingly, pharmacological targeting of purinergic signaling pathways to (re)gain control of TPC contractility represents an attractive approach for male infertility treatment or contraceptive development”.

14) Overall, the Discussion section is short and superficial. There is almost no analyses of the findings; some of the questions that should be addressed are: How results in this manuscript relates to other works in the area? Which are the unknowns that will need further investigation (gap in the knowledge)? How the purinergic system in the testis can be related to clinical applications (male fertility and/or contraception)? Which are the advances obtained in terms of imaging methodologies? How the authors think that the system is regulated: is it paracrine as they propose? If this is the case, what would be the signaling for the release of ATP? It is ok not to know the answer, it is not ok to avoid discussing these topics.

We thank reviewer 2 for making this important point, which is in agreement with a concern raised by reviewer 1 (point 7). Therefore, we have now substantially revised the Discussion. Several new paragraphs have been added to address the reviewers concerns.

Specifically, the following text has been added:

How results in this manuscript relates to other works in the area? Which are the unknowns that will need further investigation (gap in the knowledge)?

Discussion: “Among several remaining questions, future experimental efforts will have to address (i) whether TPCs are coupled by gap junctions to display coordinated activity; (ii) whether and, if so, how the final ATP metabolite adenosine affects seminiferous tubule physiology; (iii) whether Rho/Rho kinase signaling pathways modulate TPC contractility as frequently observed in other smooth muscle cells (Somlyo and Somlyo, 2003); (iv) what, if any, role is played by P2X receptor-dependent changes in membrane potential; (v) which function is served by the sustained Ca^2+^-gated Cl^–^ current (Figure 1—figure supplement 2); (vi) why periods of enhanced contractile activity are interrupted by longer quiescent episodes (Figure 7—figure supplement 2B); (vii) which additional or complementary roles in TPC physiology are played by previously proposed activators, including vasopressin, oxytocin, prostaglandins, and endothelin (Albrecht et al., 2006; Mayerhofer, 2013); (viii) whether an additional cytosolic and / or membrane Ca^2+^ oscillator (Berridge, 2008) provides an endogenous pacemaker mechanism that acts independent of purinergic stimulation; and (ix) whether, similar to vascular smooth muscle cells, some specific tone is maintained between contractions by spatial averaging of asynchronous oscillations (Berridge, 2008), a mechanism that could explain the occurrence of spontaneous low-amplitude ‘vibratory’ movements and local indentations that we (Figure 4A) and others (Ellis et al., 1981; Worley et al., 1985) have observed.”

How the purinergic system in the testis can be related to clinical applications (male fertility and/or contraception)?

Discussion: “Translation of our findings from the mouse model to humans awaits further in-depth investigation. […] Accordingly, pharmacological targeting of purinergic signaling pathways to (re)gain control of TPC contractility represents an attractive approach for male infertility treatment or contraceptive development.”

How the authors think that the system is regulated: is it paracrine as they propose? If this is the case, what would be the signaling for the release of ATP?

Discussion: “The site(s) / cellular origin of testicular ATP release as well as the mechanism(s) that trigger ATP secretion in vivo currently remain elusive. […] This way, the combined action of P2X and P2Y receptors might equip TPCs with a broader ‘two-step’ stimulus integration range.”

15) In summary, this is an excellent manuscript; however, it is written as a lab report for internal consumption. In general, the Results section needs a little more explanation on the rationale behind the experimental strategy and some clarification on the results. On the other hand, the authors should make a significant effort to improve the Introduction and Discussion section.

We thank reviewer 2 for his / her thoughtful comments, which helped us create a better manuscript. As outlined above, we have now provided the required “clarification on the results” and we made “a significant effort to improve the Introduction and Discussion section.”

Reviewer #3:[…] (1) The roles of P2X and P2Y receptors and their contribution to Ca^2+^ signaling in TPCs is somewhat confusing. In Figure 1J, the authors demonstrate that the Ca^2+^ signal in TPCs originates both from extracellular sources (P2X mediated) as well as intracellular sources (P2Y mediated). But depletion of intracellular stores (Figure 4D) seems to entirely eliminate the cellular Ca^2+^ signal, suggesting P2Y mediated signaling only? Also, given that P2Y-mediated Ca^2+^ signals do play a role, it might be worth considering moving data in Figure 1—figure supplement 1 into Figure 1 of the main paper to give them more prominence.

Reviewer 3 makes a valid point. At first glance, the role of P2X receptors – given the critical role of Ca^2+^ release from the sarcoplasmic reticulum – appears puzzling. However, it is likely that P2X receptors act as signal boosters that mediate Ca^2+^-induced Ca^2+^ release, possibly via activation of ryanodine receptors (Berridge, 2008). To clarify, we have added a new paragraph to the Discussion, which now reads:

“Excitation–contraction coupling in TPCs is poorly described. […] Therefore, it is likely that P2X receptors act as signal boosters that mediate Ca^2+^-induced Ca^2+^ release, possibly via activation of ryanodine receptors (Berridge, 2008). This way, the combined action of P2X and P2Y receptors might equip TPCs with a broader ‘two-step’ stimulus integration range.”

In addition, we acknowledge the reviewer’s argument that “it might be worth considering moving data in Figure 1—figure supplement 1 into Figure 1 of the main paper to give them more prominence.” Accordingly, we now show the relevant data (previously shown as Figure 1—figure supplement 1C–F) as new Figure 1M–P. For coherence, we now present data showing the Ca^2+^-activated Cl^–^ current as new Figure 1—figure supplement 2.

2) Figures 5 and 6 describe directional luminal transport and TPC Ca^2+^ signals respectively following focal application of ATP, with contraction occurring in a stage-dependent directional manner, while the Ca^2+^ signal is confined locally to where ATP was applied with no spread. This seems puzzling as it raises the question of what signal drives the contractions (and their directionality) in the TPCs that contract (one assumes) without an apparent Ca^2+^ signal?

Reviewer 3 raises an important point that is also brought up by reviewer 1 (point 5; see above). It is clear that we did not manage to communicate these results and, specifically, our interpretation in a clear and compelling way. We apologize and, accordingly, we have extensively revised both the corresponding Results text and the Discussion.

In Figure 5, we show that focal application of ATP triggers contractions that translate into movement of luminal content. This read-out (luminal flow change 𝑐𝑐_𝑖𝑖_) is different from the contraction measurements (flow strength 𝑠𝑠_𝑖𝑖_) conducted in Figures 3, 4 and 7. We describe the difference in the “Data analysis” subsection of the Materials and methods. We believe that this important distinction did not become sufficiently clear in the previous version of the manuscript. In a nutshell, the results shown in Figure 5 cannot be interpreted as evidence for propagating longitudinal contractions (nor do they argue against such a phenomenon). What the results *do* show is that (i) a local ATP stimulus provides the impulse to push luminal content – with or without triggering a peristaltic wave – and that (ii) for epithelia in stage group II, this movement displays directionality.

In Figure 6, we then ask if the underlying mechanism might be a peristaltic contraction, which we do assume must be causally related to TPC Ca^2+^ signals. Since we do not find evidence for a regenerative Ca^2+^ wave (Figure 6), we assume that, under the present experimental conditions, other features are involved. For example, Sertoli cell morphology could effectively cause them to function as ‘nonreturn valves’.

The absence of a regenerative Ca^2+^ wave in isolated seminiferous tubules could, of course, have several reasons, including any number of missing environmental factors present in the intact testis or simply our inability to register low-amplitude Ca^2+^ signals when using large field-of-view / low numerical aperture objectives. Importantly, we always detect coincident Ca^2+^ signals when monitoring seminiferous tubule contractions in vivo (Figure 7A and B, Video 6). Moreover, propagation of wave-like Ca^2+^ signals is also regularly observed in vivo (new Figure 7—figure supplement 2D).

In the revised manuscript, we now make an effort to describe the data and their interpretation in clear terms in the Results. Moreover, we have added dedicated paragraphs to the Discussion that pick up on the issue of Ca^2+^ signal contraction coupling and propagation. Specifically, we have revised / added the following text in the Results:

“Third, we investigated if luminal movement is restricted to the area of stimulation or, by contrast, if fluid flow propagates beyond the directly stimulated tubule section. […] These findings demonstrate directionality of sperm transport upon focal purinergic TPC stimulation in isolated seminiferous tubules.”

“As expected, ATP-induced tubule contractions also manifest as Ca^2+^ signals in TPCs (Figure 6A and B, Video 5). However, these Ca^2+^ elevations appear to be limited to those areas directly exposed to ATP (ROI 0). […] However, local contractions generate sufficient force to move luminal content beyond the directly stimulated area and, in turn, directionality of flow along short-to-medium distances (≤600 µm; Figure 5C_II_) is not critically dependent on peristaltic contractility.”

Discussion: “Whole-mount imaging of isolated seminiferous tubules reveals propagation of luminal content that extends beyond the confines of the stimulated / contracted area and displays stage-dependent directionality. […] We cannot, however, rule out that the use of large field-of-view / low numerical aperture objectives for ‘macroscopic’ imaging simply prevents the detection of low-amplitude Ca^2+^ signal spread.”

“Among several remaining questions, future experimental efforts will have to address (i) whether TPCs are coupled by gap junctions to display coordinated activity; (ii) whether… […] …(viii) whether an additional cytosolic and / or membrane Ca^2+^ oscillator (Berridge, 2008) provides an endogenous pacemaker mechanism that acts independent of purinergic stimulation; and…”

Along similar lines, what signal drives the directionality of the contractions? ATP diffusing along the tubules? Or Ca^2+^ waves spreading from cell to cell (although that should become evident in the Ca^2+^ imaging)? While this might be outside the scope of the manuscript, it might be worth addressing this issue in the currently quite focused Discussion.

Reviewer 3 emphasizes another important point. As he stated, an in-depth analysis of the mechanism(s) that govern excitation–contraction coupling and its longitudinal spread in TPCs is “outside the scope of the manuscript” and warrants systematic full analysis and an independent publication. Nonetheless, we follow the reviewer’s advice to address “this issue in the currently quite focused Discussion.” Accordingly, we have revised the Discussion as outlined in our response to concern 2 (see above). In addition, we have now added several relevant paragraphs to the Discussion that now reads:

“During spermatogenesis, apoptosis is a vital process (Print and Loveland, 2000). […] In addition, ATP release has been observed in several cell types as a result of mechanical deformation, shear stress, stretch, or osmotic swelling (Button et al., 2013) adding another putative mechanism of regenerative signaling in purinergic contraction control.”

“Notably, extracellular ATP is rapidly degraded by ecto-nucleotidases (Zimmermann et al., 2012), rendering its interstitial half-life relatively short and, thus, narrowing its paracrine radius to a few hundred micrometers (Fitz, 2007). Combined with its fast diffusion – approximately 1 μm in less than 10 ms (Khakh, 2001) – extracellular ATP bears all characteristics of a fast paracrine agent in testicular communication (Praetorius and Leipziger, 2009).”

“Excitation–contraction coupling in TPCs is poorly described. […] This way, the combined action of P2X and P2Y receptors might equip TPCs with a broader ‘two-step’ stimulus integration range.”